# Intestinal Microbiota Dysbiosis Role and Bacterial Translocation as a Factor for Septic Risk

**DOI:** 10.3390/ijms26052028

**Published:** 2025-02-26

**Authors:** Ioannis Alexandros Charitos, Salvatore Scacco, Antonella Cotoia, Francesca Castellaneta, Giorgio Castellana, Federico Pasqualotto, Maria Venneri, Angela Ferrulli, Maria Aliani, Luigi Santacroce, Mauro Carone

**Affiliations:** 1Pneumology and Respiratory Rehabilitation Unit, Istituti Clinici Scientifici Maugeri IRCCS, “Istitute” of Bari, 70124 Bari, Italy; ioannis.charitos@icsmaugeri.it (I.A.C.); giorgio.castellana@icsmaugeri.it (G.C.); federico.pasqualotto@icsmaugeri.it (F.P.); maria.aliani@icsmaugeri.it (M.A.); mauro.carone@icsmaugeri.it (M.C.); 2Doctoral School, Applied Neurosciences, University of Bari (UNIBA), 70124 Bari, Italy; 3Dipartimento di Biomedicina Traslazionale e Neuroscienze (DiBraiN), Scuola di Medicina, Università Degli Studi di Bari, Aldo Moro, 70124 Bari, Italy; salvatore.scacco@uniba.it; 4U.O. Medicina, Ospedale Mater Dei-CBH, 70125 Bari, Italy; 5Department of Intensive Care, University Hospital of Foggia, 71121 Foggia, Italy; 6U.O.C. Servizio di Immunoematologia e Medicina Trasfusionale—S.I.M.T. Ospedale Di Venere, 70131 Bari, Italy; francesca.castellaneta@gmail.com; 7Department of Public Health and Infectious Diseases, Pulmonary Division, Sapienza University of Rome, Policlinico Umberto I Hospital, Rome, Via del Policlinico 155, 00155 Rome, Italy; 8Genomics and Proteomics Laboratory, Istituti Clinici Scientifici Maugeri IRCCS, “Istitute” of Bari, 70124 Bari, Italy; maria.venneri@icsmaugeri.it (M.V.); angela.ferrulli@uniba.it (A.F.); 9Interdisciplinary Department of Medicine, Section of Microbiology and Virology, School of Medicine, The University of Bari, 70124 Bari, Italy; luigi.santacroce@uniba.it

**Keywords:** biochemistry, human microbiota, immunity, metabolome, microbiota’s crosstalk axis molecular biology, probiotics

## Abstract

The human immune system is closely linked to microbiota such as a complex symbiotic relationship during the coevolution of vertebrates and microorganisms. The transfer of microorganisms from the mother’s microbiota to the newborn begins before birth during gestation and is considered the initial phase of the intestinal microbiota (IM). The gut is an important site where microorganisms can establish colonies. The IM contains polymicrobial communities, which show complex interactions with diet and host immunity. The tendency towards dysbiosis of the intestinal microbiota is influenced by local but also extra-intestinal factors such as inflammatory processes, infections, or a septic state that can aggravate it. Pathogens could trigger an immune response, such as proinflammatory responses. In addition, changes in the host immune system also influence the intestinal community and structure with additional translocation of pathogenic and non-pathogenic bacteria. Finally, local intestinal inflammation has been found to be an important factor in the growth of pathogenic microorganisms, particularly in its role in sepsis. The aim of this article is to be able to detect the current knowledge of the mechanisms that can lead to dysbiosis of the intestinal microbiota and that can cause bacterial translocation with a risk of infection or septic state and vice versa.

## 1. Introduction

Any state of inflammation is the complex biomechanisms defence response to an acutely threatening situation, such as infection or infected tissue damage. This response of the organism may have a local or general systemic character or both [1,2].

Colonization is the establishment and multiplication of the infectious agent in the body, without causing any clinical or biological reaction. It is distinguished into normal and pathological colonization [2,3]. The presence of endogenous microorganisms can be called nonpathological colonization and is beneficial to the patient and represents the human microbiota. When the composition of the intestinal microbiota (IM) microorganisms is altered in a negative way for health, both qualitatively and quantitatively, characterized by the dysbiosis that can occur with the presence of a certain number of pathogenic microbes, then at this point we say that there is a pathological colonization [3,4].

Thus, IM plays an important role in the intestine and for all organisms because it trains the immune system to resist colonization by exogenous pathogens but also by endogenous opportunistic ones. An imbalance or sometimes destruction of the IM (for example by antibiotics, inflammation states, sepsis, and others) reduces this resistance thus facilitating the colonization of pathogens [5,6].

Some clinical situations that lead to so-called “tertiary peritonitis” or urinary tract infections may depend on the translocation of bacteria through defects in the intestinal barrier, caused by chronic diseases, local inflammatory states, or Systemic Inflammatory Response Syndrome (SIRS), sepsis o septic shock without intestinal perforations. Indeed, the uncontrolled and disturbed expression of inflammation is represented by the SIRS which can lead to septic status clinical signs (Table 1) [7].

In contrast, immunoparalysis is represented by the Compensatory Anti-inflammatory Response Syndrome (CARS). The coexistence of these two concepts leads to the formulation of Mixed Antagonist Response Syndrome, (MARS), which is reduced to an immunological disagreement and upgrades the IM dysbiosis [8].

**Table 1 ijms-26-02028-t001:** Clinical diagnostic criteria for bacteremia, SIRS, septic syndrome, severe sepsis, and septic shock. Sepsis is a rare complication of an infection, the consequences of which can be very serious and potentially fatal. It consists of an excessive inflammatory response of the body that damages tissues and organs, compromising their functioning. The diagnosis of sepsis requires the presence of at least two SIRS criteria [9].

Main Diagnostic and Clinical Criteria	
Diagnosis	Clinical/Laboratory
Bacteraemia	Positive blood cultures
SIRS	Tachypnoea (breaths per minute > 20) with or hypocapnia (PaCO2 < 32 mmHg), tachycardia (beats > 90/min), fever or hypothermia (temperature > 38.4 °C or <35.6 °C, leukocytosis (White blood cells count > 100,000 cells/µL) or leukopenia (White blood cells count < 4000 cells/μL) or immature leukocytes > 10% in peripheral blood
Sepsis	SIRS+ Bacteraemia
Septic Syndrome/Severe sepsis	*Clinical diagnosis of sepsis and one or more of the following*: Capillary reperfusion > 3 s Oliguria (urine output < 0.5/kg/h, for at least 1 h) Lactate > 2 mmol/L Sudden disturbance of level of consciousness or abnormal findings on electrocardiogram Platelets (PLT) < 100,000/µL Differing intravascular pressure (DIP) Acute respiratory distress syndrome (ARDS) Cardiac dysfunction Altered level of consciousness Systemic activation of coagulation (>dimer)
Septic Shock	Clinical diagnosis of the septic syndrome and hemodynamic disorders such as hypotension (systolic pressure < 90 mmHg or a drop of 40 mm Hg from the usual level of the specific patient’s pressure), mean arterial pressure < 60 mm Hg. Need to administer 5 µg/kg/min of norepinephrine or epinephrine to maintain blood pressure of 60 mm Hg (or >80 mm Hg in patients with a history of hypertension).
Reversible Septic Shock	Duration of hypotension <1 h and response to treatment (pressure rise)
Irreversible septic shock	Duration of hypotension >1 h, unresponsive to treatment

The microorganisms, in these cases and responsible for sepsis, are mostly coagulase-negative *Staphylococci*, *Pseudomonas* spp., *Enterococci*, *Escherichia coli*, *Clostridioides difficile*, and others. In fact, most cases of sepsis and septic shock are caused by nosocomial Gram-negative *bacilli* or Gram-positive *cocci*, rarely by *Candida* spp. or other fungi. However, it often occurs in immunocompromised patients and those with chronic and debilitating diseases [10,11].

All the above supports the possibility of a strong and close correlation between pathological colonization of the gastrointestinal tract with migration to other sites such as the lungs by the so-called microbiota’s axes such as the gut/lung and can further alter the IM which will further alter the epithelial structure of the intestine with worsening of dysbiosis [11,12].

Thus, microbial expansions may be localized (entry site or target organ) or generalized (systemic infection). Depending on the effectiveness of the defence immune system and the type of microorganism, the infection can stay local or spread to extra-local sites [13].

Microbial dispersion occurs during the inflammation of the tissues, through the lymphatics and through the thoracic duct into the blood and directly into the blood (haematogenous spread) through inflamed or necrotic vessels due to the inflammation. The infection causes damage to the human body by various mechanisms. Both the various toxins secreted by the invader and the body’s own defence mechanisms that try to prevent the infection participate in this process [14].

The aim of this narrative review research is to provide a deeper understanding of the current knowledge about the involved biomechanisms that are currently known on this topic. Therefore, it provides an overview and addresses every possible aspect of it. However, the limited research time available to explore the topic and monitor the changes that occur may be a limitation.

## 2. The Intestinal Microbiota (IM)

### 2.1. Development of the Community of Microorganisms

As we mentioned, in addition to the multitude of bacteria that surround the human body, there is also a multitude of bacteria that inhabit it, making up the so-called IM. The human body maintains a dynamic balance with its IM. The maintenance of this balance has the result that disease is not created by the bacteria of the IM and that the microbiota itself prevents the growth of pathogenic microorganisms [15,16].

The knowledge of the microorganisms that make up the microorganism’s population in the various organs is particularly important since many surgical infections can be caused by it. This knowledge contributes to the etiological-rational use of antimicrobial treatment and to the interpretation of the cultures of various biological fluids such as oral ones [16,17]. Indeed, pathological conditions that cause a rise in pH (such as high intestinal obstruction) lead to the growth of microbial strains originating from the oropharynx. On the other hand, the microbiota of both small and large intestines is abundant. However, the existence of the mucosal intestinal barrier prevents bacteria translocation [15,18,19]. Intestinal colonization begins immediately during gestation after birth and from the modality of delivery and then after birth is influenced by many factors, such as diet (breastfeeding, foreign milk), antibiotic use, age, and geographic location [20].

These microorganisms are found in the lumen, the mucosa and the outer layer of the mucus layer of the intestinal lumen. The ratio of anaerobes to aerobes is from 10:1 to 100:1 and the predominant types are the various species of the genus *Bacteroides* (66% of the total in the central colon, over 68.5% in the rectum) [21,22,23]. The existence of multiple groups of microorganisms with a high degree of degradability gives the organ its strong metabolic dynamics and helps in the recycling of nutrients. One-third of the IM is common to most people, while two-thirds is specific to each of us (Figure 1) [23,24].

Among its functions is the production of energy through the fermentation of undigested carbohydrates and the subsequent absorption of short-chain fatty acids (SCFAs) such as butyrate, acetate and propionate. It also participates in the synthesis of vitamins and in the metabolism of bile acids and sterols. In addition, it constitutes a kind of natural intestinal barrier, protecting against the invasion of foreign pathogens [25].

At the same time, the IM stimulates the regeneration of the epithelium, exerts a trophic effect on the mucous layer, and prevents the growth of a pathological microbial population. It strengthens the maturation of the immune system at the level of gut-associated lymphoid tissue (GALT) as well as acquired immunity [26,27].

The IM contributes to maintaining the normal histological structure of the intestine. Its disorders lead to intestinal atrophy and a decrease in the population of lymphocytes in the mucosa and submucosa. Whereas intestinal motility, immune factors and intraluminal secretions participate in the maintenance of the intestinal eubiosis IM [28].

The relationship between the host and IM is bidirectional (principal axis Host/IM). The host offers the microorganisms a stable environment for action and growth, as well as an energy substrate derived from nutrition and cellular degradation. Accordingly, the IM offers the host bacterial fermentation with its products. The main product, butyric acid, is the main energy substrate of the epithelial cells of the organ [29].

The main energy sources of the IM are starch and non-starch polysaccharides, the latter being better known as dietary fibre. The degradation process is gradual, non-uniform and carried out by a group of bacteria. In the long-term fight against pathogens, microorganisms have developed several pathways for their early recognition and treatment (inflammation-immune system) [30].

On the other hand, microorganisms have developed defence and attack tactics that allow them to confront, destroy, and circumvent the adversary’s antimicrobial systems. The survival of microorganisms depends on their ability to adapt and grow in specific conditions and to exploit the weak points of the host’s defences. For microorganisms to survive in a hostile environment, they must enter the body and multiply. However, during the above process, the invader stimulates the body’s defence forces [31].

In the human body, the gastrointestinal system is one of the entry gates for pathogens or microorganisms. With the ingestion, it welcomes the food as well as a multitude of microorganisms that are in the oropharyngeal cavity, permanently or occasionally. Apart from its continuous motility, the gastrointestinal system has no other special clearance mechanisms. Thus, most of them are not resistant to the acidic environment of the stomach, bile salts and gastrointestinal enzymes and thus are quickly inactivated [32]. Finally, the large intestine teems with several microorganisms, and their constant proliferation is balanced by their constant excretion in the faeces. Most of them are not resistant to the acidic environment of the stomach, bile salts and gastrointestinal enzymes and thus are quickly inactivated [33].

The human immune system consists of innate and adaptive immune responses, which have been shown to interact with the IM. The innate immune response has a key role in regulating a homeostatic environment by neutralizing pathogenic bacteria and regulating the adaptive response with the IM [34]. These effects are regulated by factors such as secretory IgA, toll-like receptor 5, autophagy and inflammation. For example, secretory IgA can bind and form complexes with commensal bacteria, which it selectively presents to dendritic cells [35].

As an anti-inflammatory molecule, secretory IgA can reduce the inflammatory response and correlate with bacterial burden in organs [36].

On the other hand, IM dysbiosis can modify the secretory of IgA and lead to dysregulated bacterial growth. Induction of IgA was confirmed as a graded response to bacterial exposure confirming the interaction between the IM and the immune system. The adaptive immune response is another important part of maintaining a healthy microbiome and immune balance [37].

Especially the adaptive immune response is achieved by the differentiation and maturation of B and T cells and the establishment of immune tolerance to the IM. The gastrointestinal tract contains many immune cells, which constantly communicate with the IM. The maturation of the immune system requires the development of common microorganisms. One of the mechanisms of the IM influencing the immune system is the regulation of neutrophil migration, which in turn affects the differentiation of T cells into various types such as T helper cells and T regulatory cells [37].

Disturbance in the growth of microorganisms during the maturation of the immune system could lead to altered immune tolerance and autoimmune diseases. Additionally, heterogeneous molecules produced by microflora can induce an immune response and stimulate inflammation or chronic tissue damage [38]. A key role mechanism of antimicrobial protection involves the mucosa in the colon, which acts as a barrier, keeping luminal microbes away from the epithelium. This protective mucus layer, secreted by intestinal cells, is composed of mucosal glycoproteins that extend up to 150 μm from the colonic epithelium. The denser inner layer remains free of organisms, while the dynamic outer layer serves as a source of glycans, providing food for microbes [39,40].

In contrast, the small intestine has a discontinuous and less robust mucus layer, making antimicrobial proteins more efficient for the defence. The IM, through its constituents and metabolites, stimulates the synthesis of antimicrobial proteins, such as cathelicidins, C-type lectins and (pro)defensins by host Paneth cells, mainly found in the cecum and colon [41]. These molecules activate specialized molecular signalling pathways that are necessary to maintain the integrity of the mucosal barrier and promote the production of antimicrobial proteins, mucosal glycoproteins, and IgA antibodies [41,42].

In addition, the IM uses another defence mechanism to regulate the overgrowth of pathogenic strains by inducing the production of local immunoglobulins. Gram-negative bacteria (such as especially *Bacteroides*), stimulate intestinal dendritic cells, inducing the expression of secretory IgA (sIgA) by intestinal mucosal plasma cells [43]. The sIgA coats the IM, rendering it resistant to degradation by bacterial proteases, while also preventing the transport of gut microbes into the circulation, thus limiting the systemic immune response. This complex interplay between the IM and the mucosal immune system highlights the complexity of maintaining gut health [44].

### 2.2. The Intestinal Microbiotas’ Dysbiosis

IM dysbiosis is an imbalance or disorder in the composition and function of the IM. It can be characterized either by a loss of overall microbial diversity or by transient or permanent changes in the composition of the IM. The IM of a healthy individual consists of a diverse array of bacteria, viruses (viroma), fungi, and other microorganisms (protozoa such as *Entamoeba histolytica*) that play a critical role in maintaining the overall health of the host [45]. Essentially, the term dysbiosis reflects a disturbance in the normal homeostasis of the IM, which is characterized by an imbalance in bacterial composition, changes in metabolic activity and changes in bacterial distribution in the intestine [46]. There are three types of dysbiosis by the microorganisms: (a) loss of beneficial bacteria, (b) overgrowth of potentially pathogenic bacteria, and (c) loss of bacterial diversity. In most cases, these three types coexist, resulting in an increase in potentially pathogenic bacteria and a decrease in beneficial bacteria [47,48].

Despite the diversity of intestinal microorganisms, research shows that in most individuals the IM can be divided into three enterotype categories according to the bacteria prevalence; (1) *Bacteroides*, (2) *Prevotella* and (3) *Ruminococcus*. Thus, each enterotype is dominated by the phyla of bacteria from which it takes its name [49].

The first enterotype shows an increased ability to produce energy from carbohydrates and proteins, the second can degrade glycoproteins found in the intestinal mucosa, while the third is related to the degradation of mucin and the membrane transport of sugars. At the same time, all three enterotypes perform metabolic functions and synthesize various vitamins [50].

However, the above separation is intended to simplify the analysis of the human IM as there are many different cases that do not belong to any of these categories. The composition of microbial populations differs and changes in everyone and depends largely on their diet and other factors [51].

In general, the microorganisms that make up the human IM belong to approximately 500 species and the bacteria that live in the human body belong to 1000–1150 different species [52,53].

In a healthy IM, there are mainly the species from the phyla *Bacillota* and *Bacteroidota*, but also *Actinomycetota* and *Deltaproteobacteria* in smaller numbers. However, this is not certain, as variability has also been observed in healthy individuals. As we mentioned, in addition to these bacteria, primary pathogens are also present in the gastrointestinal system in high or low concentrations, such as *E. coli* and “*Salmonella enterica*” responsible for infections and sometimes sepsis [54].

A healthy IM appears to have a reduced ratio between *Bacillota*/*Bacteroidota*, while at the same time, it consists of *Peudomonota* pathogenic bacteria and bacteria that increase the production of the butyric acid metabolite. This implies good health of the individual, but also the prevention of various diseases [50]. Thus, the main functions of the IM include also the metabolome (metabolic activities) that participates in energy production and storage, nutrient absorption, host defence against invading pathogens, trophic effects on the intestinal epithelium, homeostasis of the intestinal epithelium, and immune regulation responses [55,56].

Establishing a “healthy” relationship between the microorganisms and the gastrointestinal tract early in life appears to be critical for maintaining gut homeostasis and preventing the onset of pathogenic conditions and infections that can lead to serious infections and sepsis status [55,57].

Dysbiosis occurs when there is a shift in the normal balance of these microorganisms. This imbalance can lead to a reduction or loss of beneficial bacteria and an increase in populations of potentially harmful bacteria. This disruption can lead to changes in the diversity, abundance and overall structure of the IM [58].

The changes in the IM can occur following exposure to a variety of environmental factors, including diet, antibiotics, infections, SIRS, septic shock, xenobiotics (such as bisphenol A, glyphosate, air pollutants, polychlorinated biphenyls (PCBs), heavy metals). However, the host’s IM can facilitate the metabolism of certain drugs. For example, digoxin, which belongs to the cardiac action glycosides, increases the levels of a cytochrome. This can be inhibited thanks to the action of the *Eggerthella lenta* (*Actinomycetota* phyla), and it is so it can be converted in 10% of cases to dihydrodigoxin, an inactive metabolite. In addition, the bacterial metabolite p-cresol by *E. lenta* has a positive effect on the paracetamol’s metabolism, due to the competitive inhibition of hepatic sulfotransferases [59,60].

## 3. Intestinal Bacteria Translocation Risk

The functions of the intestinal wall include the absorption of nutrients. The transport of substances from the intestinal lumen to the inside of the body takes place through the epithelial cells (intercellular) and through the occlusive compounds that connect them (paracellular) [61].

Intestinal bacterial translocation is defined as the phenomenon in which bacteria, their derivatives (metabolites and conjugates) or both cross the intestinal barrier and colonize extraintestinal tissues [62]. Bacterial translocation can be a normal phenomenon that also occurs in healthy individuals without consequences. But when the immune system is challenged extensively it fails and results in septic complications that are localized in sites far from the main “focus”. Translocation is always in some way a consequence of an interruption of the gastrointestinal barrier, following functional and/or anatomical damage, thus supporting the hypothesis of “intestinal failure” [61,63]. The mesenteric lymph nodes are the first extraintestinal organ to colonize, followed by the liver, spleen, and systemic circulation. There are three main mechanisms that promote bacterial translocation, the microbiota’s overgrowth, the conditions of immunosuppression and the damage to the architecture of the intestinal mucosa [63]. When one of these mechanisms is particularly severe, its action is prolonged or combined with another, it increases the severity of intestinal bacterial translocation and can lead to severe sepsis, Multiple Organ Dysfunction Syndrome (MODS), and exitus. Several studies have shown that microorganisms that most often participate in the intestinal bacterial translocation process are *Candida* spp., *Escherichia coli*, *Klebsiella pneumoniae* and *Proteus mirabilis* and can create infections and sepsis in other apparatus such as the urinary tract [64].

Anaerobic bacteria, although they constitute most of the intraluminal microbiota’s microorganisms, have not been described to be associated with intestinal bacterial translocation in the blood, possibly due to the presence of oxygen in the blood which inhibits their growth. Several studies have shown that intestinal epithelial apoptosis is caused by various pathological conditions such as coeliac disease, Crohn’s disease, ulcerative colitis, etc. [65,66]. Indeed, increased apoptosis is observed in natural mucosal injury and leads to epithelial atrophy, mucosal intestinal barrier dysfunction, and increased intestinal permeability. Also, various conditions cause an increase in apoptosis, such as infections by several bacteria (such as *Salmonella* spp., *E. coli*, *Shigella-flexneri*, *C. difficile*), ischemia-reperfusion, idiopathic inflammatory bowel disease, radiations, nonsteroidal anti-inflammatory drugs, cytostatic drugs and other [67].

### 3.1. Factors That Limit Intestinal Damage and Intestinal Bacterial Translocation

In the long-term fight against microbes, animal organisms have developed highly efficient systems for their early recognition and defense in the long-term fight against microbes, highly efficient systems for their early recognition and defense have been developed in our organism, through the immune system. On the other hand, microorganisms have developed defense and attack tactics that allow them to confront and circumvent the adversary’s antimicrobial systems. The survival of microorganisms depends on their ability to adapt and grow in specific conditions and to exploit the weak points of the host’s defenses [68].

For microorganisms to survive in a hostile environment, they must enter the body and multiply. However, during the above process, the invader stimulates the body’s defense forces. In the human body, the gastrointestinal system is one of the entry gates for microbes [69]. With the ingestion, it welcomes the food as well as a multitude of microbes that are in the oropharyngeal cavity, permanently or occasionally as oral microbiota. Apart from its continuous motility, gastrointestinal systems haven’t any other special clearance mechanisms [15,69].

From studies mainly in experimental animal models, it has emerged that there are factors that enhance intestinal barriers and can limit bacterial translocation. These include the enteral nutrition of glutamine and arginine, trophic factors (such as growth hormones, neurotensin, bombesin, insulin-like factor I, and epidermal growth factor), antibiotics and substances such as lactulose. Enteral nutrition significantly limits the atrophy of the intestinal mucosa and improves local immunity, reducing bacterial translocation [70,71,72,73,74,75,76]. In a previous study on rates, the combination of glutamine and growth hormone reduced bacterial translocation, because it can be affecting the mucosal cells by bio-mechanisms related to nutrient delivery and by increasing s-IgA secretion [71]. Arginine improves the local immune reaction of the intestinal mucosa [72]. The positive effect of neurotensin on preventing bacterial translocation and preserving intestinal mucosal integrity after abdominal radiation was studied in rats [74]. A prospective, multigroup trial in which animals (outbred ICR mice) fed each test diet were randomized to receive either bombesin or saline for seven days demonstrated the positive effects against bacteria translocation [75]. Some studies report satisfactory attempts to modulate IM, with the aim of limiting it by administering poorly absorbable antibiotics (such as rifaximin) and lactulose that bind microbes and endotoxins, or by replacing it with “protective” bacteria (probiotics), by administering preparations with a high content of species from the *Lactobacillaceae* family, but their usefulness in clinical practice has not been established. Indeed, rifaximin demonstrated to be effective in reducing bacterial translocation to the colon in mice with induced colitis [70,76].

### 3.2. Factors That Damage Mucosal Intestinal Barrier and Promote Bacterial Translocation and Septic Risk

#### 3.2.1. Not Affecting Paracellular Permeability

The gastrointestinal barrier has extrinsic and intrinsic defence components: the extrinsic barrier, located inside the lumen, stabilizes the entire intestinal material on the epithelial surface and consists of mucus, secretory IgA and intraluminal bacterial community; the intrinsic components are represented by epithelial cells and the space around them, further protected by the presence of tight junctions (occluding junctions). These components of the gastrointestinal barrier can be overcome in case of functional and anatomical alteration of the mucosa [77].

Several factors can damage the mucosal intestinal barrier and lead to bacterial translocation such as marked malnutrition, atrophy of the intestinal mucosa, hypoproteinaemia, immunosuppression and others [78]. Intestinal diseases (such as inflammatory bowel disease, intestinal obstruction, intestinal ischemia, liver cirrhosis, severe pancreatitis and others) and local intestinal endotoxemia can be associated with increased bacterial translocation due to mucosal intestinal barrier damage [79]. For example, inflammatory bowel disease can lead to complications, such as erosion and perforation of the intestine, which can lead to infections (such as peritonitis) and lead to sepsis. The risk of SIRS and septic status can be caused also by severe surgical or not conditions (transplants, burns, severe trauma, haemorrhagic shock, and others) causing damage to the mucosal intestinal barrier and an increase in its permeability [80].

On the other side severe dysbiosis of the IM can also lead to local and systemic endotoxemia which, through the production of nitric oxide (NO), leads to a deterioration of the functionality of the intestinal mucosal barrier, therefore facilitating bacterial displacement and increasing the risk of an infection which can evolve into a septic state [81,82].

The parenteral nutrition after long-term administration seems to lead the intestinal mucosa to a state of “starvation” and eventually to atrophy. We can have not only local dysbiosis inflammation but also bacterial translocation and distant infections such as in the respiratory system (pneumonia and others) [83,84]. Therefore, parenteral nutrition deprives the body of the numerous nutritional benefits that come from enteral nutrition. For this reason, it is recommended to consider at least the partial possibility of enteral nutrition where possible [85].

Finally, the radiation damage causes immediate destruction of the intestinal epithelium and a significant degree of bacterial translocation in laboratory animals and humans. Indeed, local ablation of tumours using *Radiofrequency ablation* (*RFA*) appears from experimental studies in rats that after the treatment, the mucosal intestinal barrier is impaired resulting in the induction of endotoxemia [86]. Indeed, in an experimental study, Wistar rats were used and subjected to RFA for 28.5% of the liver volume. It was noted that endotoxin levels increased in both the portal and systemic circulation and the ileal mucosa gradually became atrophic, with a decrease in the glutathione/glutathione disulfate (GSH/GSSG) ratio. Hepatic RFA led to endotoxemia and translocation of intestinal bacteria to proximal and distal organs [86,87,88].

#### 3.2.2. Affecting Paracellular Permeability

Paracellular permeability, as mentioned above, is the main route of transport of substances through the mucosal intestinal barrier. The paracellular pathway is built by the tight junctions’ compounds (a complex combination of transmembrane integral proteins, including claudins, occludin, and junctional adhesion molecules), which are in a dynamic equilibrium. This balance is regulated by both extracellular and intracellular events. Intracellular events that can affect the stability of occlusive compounds are related to the energy homeostasis of the cell. This in turn is expressed by changes in cAMP levels and energy depletion [89,90]. A decrease in ATP levels causes a decrease in the expression of the tight junctions’ compounds, while on the contrary, a decrease in cAMP causes an increase in transepithelial resistance and a decrease in paracellular permeability [91].

The morphology of the tight junctions’ compounds is affected depending on the differentiation of the cell within the cell cycle. This regulation by the cell cycle can be both quantitative (increased or decreased expression of some molecular components) and qualitative (different expression of claudins during development) [92]. Various substances with regulatory action on the mucosal intestinal barrier through paracellular permeability (Figure 2) [92,93]. Proteases are substances secreted by leukocytes that cause the destruction of tight junction compounds and fragmentation of their macromolecular components [94]. Interleukins, depending on their type and the specific tissue where they act, can either strengthen or disrupt the tight junctions’ compounds [95]. Interferons exert both positive and negative effects on tight junctions in different epithelia in ways analogous to interleukins [96]. D4 leukotrienes cause reorganization of the action network and therefore affect the occlusive junctions. Glucocorticoids cause disruption of the epithelial barrier through phosphorylation of serine/threonine sites [97]. IgM and IgG immunoglobulins interact with coxsackievirus and adenovirus receptor molecules at sites of inflammation, which probably play a role in polymorphonuclear migration. Growth factors act on epithelia and are associated with a reduction in barrier function and disorganization of occlusive compounds [98]. Fibroblast growth factor (FGF), hepatocyte growth factor (HGF) and vascular endothelial growth factor (VEGF) cause an increase in paracellular permeability. Of course, other growth factors may have the opposite effects [99]. Oxidative stress causes an increase in paracellular permeability through the phosphorylation of annexins at tyrosine sites. This phosphorylation induces the release of annexins from the blocking compounds and their intracellular movement. Disturbance of calcium balance appears to be associated with barrier disruption. More specifically, the decrease in calcium levels is associated with barrier disruption and an increase in paracellular permeability [100,101].

Extracellular events that may have a regulatory role in junctional structure and function include both direct and indirect interactions of junctional proteins with other cellular proteins [102].

A typical example of direct bidirectional action is the interaction of leukocyte membrane antigens with the endothelium, which causes local relaxation of occlusive junctions [102,103]. Also, direct interactions with antigens of the extracellular space can have a regulatory role, such as for example the interaction of claudins 3 and 4 and annexins with the enterotoxin of *Clostridium perfringens*, for which they are receptors. The category of extracellular events with an indirect regulatory role in the structure and function of the occlusive junctions includes the indirect effect of cytokines and hormonal stimuli on the mucosal intestinal barrier [104]. Furthermore, it has been shown that upregulation of Claudin-2 can increase intestinal permeability, leading to immune activation, dysbiosis and mortality in sepsis [89,104].

The mechanisms through which the various stimuli affect the structure and function of the occlusive compounds are the subject of research activity. Phosphorylation is likely a common, but not unique, regulatory mechanism. Phosphorylation occurs at serine/threonine and tyrosine sites [105]. Depending on the position carried out, opposite results appear. Thus, a high degree of serine/threonine phosphorylation is observed when the integrity and morphological stability of the occlusive junctions are maximal, while a low degree of phosphorylation at the specific sites causes disorganization of the occlusive junctions and cytoplasmic localization of annexins [106,107].

On the other hand, phosphorylation at tyrosine sites causes disorganization of the occlusive compounds and cytoplasmic localization of annexins which also loses its ability to bind to ZO-1, ZO-2 and ZO-3 [108]. Various protein kinases or phosphatases have been found to be directly or indirectly associated with the intracellular portion of occlusive compounds. These include protein phosphatase 2A (PP2A), protein kinase C (PKC), casein kinase 2 (CK2), phosphatidylinositol kinase 3 (PI3K), cAMP-dependent kinase, tyrosine kinase, and tyrosine phosphatase [109].

Another post-translational mechanism that may be involved in the regulation of occlusive compounds is the N-glycosylation that the CAR T cell molecule undergoes [110].

Also, annexins undergo proteolytic fragmentation during polymorphonuclear migration through the paracellular pathway, resulting in the formation of a 22 kD molecular weight product incapable of participating in the paracellular barrier. In conclusion, as shown by the data to date, the structure and function of barrier junctions can be regulated through transcriptional and translational modification [111].

## 4. Role of Sepsis in Promoting Intestinal Bacterial Translocation

As we mentioned, bacterial translocation is the invasion of viable and non-viable germs and their bioproducts through the intestinal mucosa to the mesenteric lymph nodes, spleen, liver, and peritoneum [112].

It frequently occurs in patients with intestinal obstruction and is the cause of subsequent sepsis, but, as demonstrated by many experimental studies, with each type of “injury” the intestine itself becomes a “target organ” and its dysfunction leads to the alteration of its permeability, facilitating the phenomena of bacterial translocation and absorption of exotoxins, endotoxins and other debris [112,113].

In fact, in a prospective study cultures of nasogastric aspirates from 279 surgical patients were developed and compared with cultures from mesenteric lymph nodes along with cultures of subsequent septic complications, it was noted that proximal intestinal colonization was associated with both greater bacterial translocation and greater septic morbidity [114,115].

During an endotoxemia state may be modelling the hyperinflammation associated with early sepsis. Indeed, lipopolysaccharide (LPS) from Gram-negative pathogens activates the immune system, leading to hyperinflammation with microcirculatory outcomes. In addition, other toxins would be Gram-positive peptidoglycan and lipoteichoic acid. Thus, these toxins activate various signalling pathways inducing a modelling of sepsis playing a fundamental role in its pathogenesis and consequently in the risk of bacterial translocation [116,117]. Enterotoxins (exotoxins) are protein structures released by a microorganism that affect the intestine. They are heat labile (>60 °C), low molecular weight and water-soluble. Enterotoxins are often cytotoxic by altering the permeability of the apical membrane of the mucosal cells of the intestinal wall and causing cell death [118].

Their action leads to an increase in the intestinal permeability of the membrane for chloride ions through the formation of pores driven by increased cAMP or increased intracellular calcium ion concentration. Several microbial organisms contain the enterotoxin necessary to create this effect, such as *C. difficile*, *C. perfringens*, *Vibrio cholerae*, *Staphylococcus aureus* (enterotoxin B), *Yersinia enterocolitis*, *Shigella dysenteriae* (Shiga toxin). It has been noticed that the purified Shiga toxin has a lethal toxicity to mice when injected intraperitoneally with an LD50 of 28 ng per mouse, instead, *C. difficile toxin* B (TcdB) LD50 in mice is 20 ng (1 μg kg^−1^), with 100 ng causing acute severe damage in all animals [118,119,120]. Also, *Reoviridae* (*Rotavirus*), *Caliciviridae* (*Norovirus*) and *Astroviridae* have been found to contain an enterotoxin NSP4 which by a presumed pathway of toxicity is that activates the increase in cellular calcium concentration and subsequent secretion of chloride from the cell. Normal osmotic pressures are altered, and this prevents the absorption of water, causing diarrhoea and can contribute to bacterial translocation [121,122,123].

Therefore, the factors that can stimulate bacterial translocation are not only immunodeficiencies, immunosuppression, and dysbiosis from non-infectious causes but also infectious ones, including generalized inflammatory states such as SIRS and sepsis that alter the permeability of the mucosal barrier and the patient’s immunity functions [124,125].

It occurs through transcellular and paracellular passages and can be measured both directly by mesenteric lymph node culture and indirectly by peripheral blood cultures, detection of microbial DNA or endotoxins and urinary excretion of unmetabolized sugars [126,127,128].

The relationship between bacterial translocation and MODS is therefore in the fact that, once the hepatic filter and subsequently the pulmonary filter have been passed, there can be diffusion into the bloodstream with consequent bacteraemia, fungoides and endotoxemia which cause (and maintain) systemic sepsis and therefore MODS [129,130].

On the other hand, in critically ill patients with severe alteration of homeostasis, many factors predispose to bacterial and endotoxin translocation, both in conditions of unaltered anatomical barrier and alteration of the intestinal mucosa [131].

It has been noted that the translocation of small intestine local amounts of bacteria endotoxins enhances the endoplasmic reticulum stress and especially the Kupffer cells in the liver. Thus, a malfunction of both the mucosal barrier and endoplasmic reticulum stress leads to systemic endotoxemia, and lead to further mucosal barrier damage downregulation of immune system function, coagulation system, Kupffer cells, gut microbiota axes and organ dysfunction. Thus, these conditions can lead to more bacterial translocation [132,133].

The modifications of the intestine therefore, induced by surgical manoeuvres such as laparotomies, through the same procedures of anaesthesia, and as mentioned the same states of shock, through the ischemia/reperfusion mechanism and neuro-endocrine reflexes, determine early dynamic ileus [134,135].

Furthermore, treatments in Intensive Care Units and the “non-use” of the gastrointestinal tract for parenteral nutrition, promote further dysfunctions in the gastrointestinal tract itself, leading to a picture of progressive ileus, colonization of the upper digestive tract and the reduction of the function of the entire lymphoid tissue present in the gastrointestinal tract [136,137].

The upper tract acts as a “reservoir” of pathogens, while the local and systemic defence mechanisms that prevent bacterial spread undergo profound alterations. In practice, the gastrointestinal tract cannot be considered a passive organ, but a dynamic organ, whose function covers the traditional role of nutrient absorption together with a defence activity against the potential harmfulness of the IM [138].

In critically ill patients, we can find a significant increase in the upper digestive tract IM with pathogen bacteria that can contribute significantly to the onset of intra-hospital infections [139].

In fact, over 90% of subjects with hospital infections have as their cause a microorganism simultaneously present in the “upper” gastrointestinal tract. Although not fully demonstrated, selective decontamination of the gastrointestinal tract and/or the use of cytoprotective drugs for the prevention of stress ulcers seem to reduce the incidence of nosocomial infections [140].

The IM plays an important role in the intestine’s resistance to colonization by exogenous pathogens, the so-called “colonization resistance”. The reduction or sometimes destruction of the IM by antibiotics reduces this resistant colonization of exogenous pathogens. Some situations that lead to “tertiary peritonitis” may depend on the passage of bacteria through defects in the intestinal barrier, while in a few patients, there is evident communication between the intestinal lumen and the intestinal barrier [141]. Once bacterial translocation has occurred, the development of a consolidated systemic septic state and the onset of MODS would depend on several factors: (a) bacterial load and virulence, (b) duration and frequency of bacterial translocation events, (c) bacterial “clearance” capacity of the liver and lung (d) overall systemic capacity of the immune response and the ratio between pro-inflammatory mediators compared to inflammatory mediators [142]. These conditions act in synergy; therefore, ultimately the event of bacterial translocation should be considered the result of multiple physiological and pathological events. For example, if there is an increase in dysbiosis of the IM, it will also be possible that of the microbiota of the lower airways which, through the intestine/lung axis, will allow a probable translocation of pathogenic or opportunistic bacteria [143,144]. Finally, a more plausible explanation is that bacterial translocation is the late manifestation of the MODS, but not its, “principal unhealthy promoter” and is one of the causes of bacterial translocation [145]. Furthermore, this condition of bacterial translocation can cause in the first moment ARDS that can progress to a MODS. Indeed, the hypothesis of the involvement of the lung, which is the first organ to damage, is based on the consideration that the lymph coming from the mesentery reaches the subclavian vein through the thoracic duct, thus reaching the heart and the lung directly [145,146]. This consideration provides the identification of “MODS inducing factors” Indeed, it seems that the evolution towards MODS is probably also due to intestinal lesions in patients undergoing major abdominal surgery, with systemic diffusion of non-microbial but tissue-damaging factors through intestinal lymphatics. These observations have led to the gut-lymph hypothesis of MODS that increases the probability of bacterial translocation [146].

## 5. Gut-Brain Axis Role in Sepsis

The IM is a biochemical “factory” in which the host has a bi-directional relationship condition with and communicates with other organs, regulating their functionality in certain aspects. Therefore, there are IM “bio humoral connection axes” [147]. Among the various axes, the most studied are intestine-liver, intestine-kidney, intestine-lung, intestine-heart, intestine-urogenital system and gut-brain they are strongly linked by a condition of eubiosis or dysbiosis of the microbiota. Here we must mention the pneumogastric nerve or vagus nerve in the intestinal brain axis. In fact, it is mainly through vagal innervation that the brain regulates intestinal motility, and visceral responses to pain, fear, anxiety and apprehension [148]. Therefore, not only does the brain influence the intestine, but also the IM can influence the proper functioning of the brain and its development over time. It has now been noted that some bacteria in the IM are able to produce neurotransmitters such as serotonin, polyamines, dopamine GABA and others [149]. The concept of the gut-brain-microbiota axis (GBA) implies that the gut microbiota interacts with and influences the nervous system. The interaction is bidirectional. The central nervous system and enteric nervous system (ENS) communicate with the gut microbial population through neural, endocrine, immune, and humoral connections that influence their composition and behaviour. In agreement with the above, agranulocytosis of a segment of the intestine can lead to an abnormal distribution of the IM [150]. Dysbiosis of IM is involved in neuroinflammation of patients with sepsis, its mechanism may include increased permeability of the intestinal wall, and this negatively affects the GBA axis and can lead to bacterial translocation [151].

Furthermore, the wide inflammation state that occurs during severe sepsis is strongly associated with patients who may experience impairments in memory, concentration, verbal fluency, and executive functions during and after the onset of the disease. This occurs because sepsis affects the homeostasis of the GBA, which leads to neuromodulation and immune alteration, neurological dysfunction, and therefore cognitive deterioration. Indeed, in an animal study with adult Wistar rats subjected to surgical procedures involving cecal ligation and perforation (CLP) or sham (non-CLP) this very link between sepsis and neurodevelopmental deterioration was noted [152].

In an animal study (mice) it was considered that septic patients with acute kidney injury and common metabolic disorders are more susceptible to sepsis-associated encephalopathy [153]. In fact, several studies have indicated that susceptibility to MODS also affects the CNS and could be modulated by IM [154,155]. In addition, the metabolome by microbial bacteria acts with functional effects on the GBA. They studied a metabolite, indole-3-propionic acid, for its strong neuroprotective action suggesting that the gut-brain axis is an upstream regulator of sepsis-associated encephalopathy [156]. Furthermore, they also considered the variability in sepsis-induced intestinal dysbiosis mediates differential susceptibility to sepsis-associated encephalopathy in mice with experimental sepsis induced by cecal ligation and puncture (*CLP*). They noted that there are potential mechanisms through which IM mediates this susceptibility to sepsis-associated encephalopathy and during the progression of sepsis [146,157].

## 6. Assessment of Intestinal Permeability as a Strategy to Strengthen Intestinal Barrier Function to Avoid Bacterial Translocation

Intestinal permeability and integrity can be measured in many ways; in vivo or in vitro techniques can be used, animal or human models can be used, and different types of molecules can be used for measurement (ions, carbohydrates of different molecular weights, macromolecules and antigens, bacteria or their products) and finally, the sampling site can vary (peripheral blood, urine, feces) [158]. Functionally, however, the tests useful for evaluating intestinal permeability can be divided into two large groups, the first includes in vivo and ex vivo tests that directly study intestinal functionality by evaluating the passage of certain substances through it (e.g., lactulose/mannitol test, PEG) or the presence of substances related to intestinal commensal bacteria; the second group includes tests in which substances are measured that have been identified as markers of damage to the intestinal barrier, therefore these are biomarkers or histological markers such as Intestinal Fatty Acid Binding Protein (I-FABP), citrulline, and faecal calprotectin) [159]. Each of these tests has a role in the evaluation of diseases related to the alteration of intestinal permeability, evaluating different aspects and in different moments during the pathological status, but all can present limits of feasibility, therefore it is important to know the main differences and indications to be able to draw useful information [160].

The Lactulose/mannitol test most of all uses large-sized oligosaccharides, such as lactulose, combined with sugars of reduced molecular weight such as mannitol. These substances are administered orally and will then be sampled in the urine. This test can only be used to evaluate permeability at the small intestine level because lactulose is degraded by bacteria in the large intestine [161,162]. To study the entire intestine, substances that are not degraded can be used, such as sucralose, which is added to the previous multisugar (sucrose, lactulose, mannitol, and sucralose) test (MST), thus providing a triple test. The molecules of high Mm cross the intestinal barrier passing into the intercellular spaces only if its function is altered, i.e., due to alterations of the tight junction, and flow into the circulatory stream to be excreted finally at the renal level [163]. The smaller molecules, instead physiologically freely cross the barrier through small pores present in all cells. In case of alteration of permeability, these pores atrophy, and their passage is reduced. When administered, lactulose can determine an increase in intestinal motility. An increase in permeability for sugars has been demonstrated not only in chronic intestinal diseases but also in critical patients and in patients undergoing major surgery [164]. However, some studies have highlighted that the study of intestinal permeability with this method in critical patients may hide some pitfalls; first, the alterations of intestinal motility and the altered clearance of different sugars are factors that are complicated to evaluate in these patients. Secondly, the use of mannitol is not suitable for patients who are transfused [165].

The LPS dosing in critical patients has been successfully used to highlight endotoxemia, especially in septic patients, although high levels of LPS have also been found in obese patients and patients with metabolic syndrome. Plasma LPS detection indicates bacterial translocation from the intestinal lumen to the bloodstream because of a deficit in intestinal barrier function [166,167].

The limit of this test lies in the measurement and standardization of the test, which is still far from being used in clinical practice. As an alternative to the study of endotoxemia, circulating levels of EndoCab (Endotoxin core antibody) can be measured, a test that can assess acute intestinal barrier damage. Most studies demonstrate a reduction in circulating levels of this antibody in the postoperative period in accordance with the degree of exposure to endotoxin [168].

Thus, the consumption of this circulating immunoglobulin following the translocation of endotoxin through the intestine can be used for the indirect evaluation of the intestinal barrier function [169].

This approach is successful only in patients with acute pathology while it is rarely used in chronic patients. Another interesting aspect is that the translocation of LPS through the intestinal barrier and reaching the liver via the bloodstream determines hepatic inflammation with consequent liver steatosis [170].

The Citrullinemia dosage is the study of intestinal functionality and can then be conducted by analyzing the presence of markers indicating intestinal damage [171]. Citrulline is an amino acid that is not present in proteins but is produced by the small intestine starting from glutamine and can be taken as an index of the functionality of the enterocyte mass at this level [172]. A loss of epithelial cells of the small intestine determines an increase in permeability and a reduction in blood levels of citrulline. It has been demonstrated that this process can occur during hematopoietic transplants, and in pediatric patients following chemotherapy. The specificity and sensitivity of this test have been shown to be greater than sugar permeability tests [173,174].

Fatty acid binding proteins (FABPs) dosage is another marker of intestinal damage. Those are small water-soluble proteins present on the surface of mature enterocytes throughout the intestine; their function is to transport fatty acids from the apical membrane of enterocytes into the endoplasmic reticulum where the biosynthesis of complex lipids occurs [175]. FABPs can be measured both at plasma and urinary levels with the ELISA technique. The detection of basal levels of FABPs reflects the physiological turnover of enterocytes, on the contrary, high levels indicate damage to the intestinal epithelium. High circulating and urinary levels of FABPs have been described in patients with intestinal ischemia, in cases of SIRS, and in necrotizing enterocolitis. Furthermore, FABPs have been used as a marker of intestinal barrier alteration in liver transplant patients, with important prognostic implications [176]. The test is currently used to monitor the progress of chronic intestinal disorders and/or diseases and metabolic syndromes [177].

The Fecal calprotectin dosage is a highly promising marker is fecal calprotectin. In fact, it is very resistant to proteolysis if kept at room temperature for more than a week. Calprotectin is released during cell activation and death and has antiproliferative, antibacterial and immunological functions. It is currently used in clinical practice to evaluate the progress of IBD [178].

## 7. The Management of Dysbiosis During Sepsis: Functional Foods and Fecal Microbiota Transplant

Macronutrients, including carbohydrates, proteins, and fats, several micronutrients, probiotics, polyphenols, and phytochemicals influence the composition and diversity of the host EM. The effects of dietary protein on IM have been extensively studied. Lower numbers of *Bifidobacterium adolescentis* and higher numbers of *Bacteroides* spp. and *Clostridia* spp. have been found in individuals consuming a high-beef diet compared to non-meat eaters [179].

The effect of dietary protein on IM has also been studied in individuals consuming different forms of protein, such as animal protein from meat, eggs and cheese, whey protein or vegetarian protein sources (pea protein) [180].

Most studies have shown that protein consumption is positively associated with overall GM diversity during septic status. Furthermore, in sepsis after the acute phase, the patient must receive an increase of proteins around 1.2–2.0 g/kg/day to minimize further loss of lean mass and to promote early mobilization [181].

Diets high in saturated and trans fatty acids are detrimental to health and can lead to dysbiosis. A diet high in saturated fatty acids increases the relative proportion of *Faecalibacterium prausnitzii* [182]. However, beneficial fatty acids, such as monounsaturated and polyunsaturated fatty acids, play a crucial role in reducing the risk of chronic diseases and infectious status. Indeed, several studies mention the beneficial effects of these pathologic conditions. In a systematic review and meta-analysis of randomized trials, it has been revealed that Omega-3 fatty acid supplementation might be associated with reduced mortality in patients with sepsis [183].

The key role of probiotics focused on improving intestinal health, ameliorating symptoms related to lactose intolerance and reducing the risk of various gastrointestinal diseases is increasingly recognized. In addition, probiotics have become promising modulators of immune system function, effectively enhancing both innate and adaptive immunity, promoting changes in the composition and activity of the IM microorganism’s profile, enhancing intestinal barrier integrity, and synthesizing first-line antimicrobial compound lines to protect against attacking pathogenic microorganisms [184]. Thus, they have an immunoregulatory action even in cases of infections and septic state with the production of metabolites (metabolome), control transport through the intestinal barrier, correct dysbiosis and lead to an increase in cellular turnover and competition with pathogens [185]. Although some probiotic strains, such as *Bifidobacterium* spp. *Saccharomyces* and *Lactobacillaceae* family, are known for their proven safety and efficacy, there is increasing interest in investigating the therapeutic abilities of lesser-known strains, such as *Roseburia* spp. *Akkermansia muciniphila* spp. and *Faecalibacterium* spp., which hold great promise as future additions to the probiotic composition [186,187]. In particular, the scope of functional foods (such as probiotics, prebiotics, postbiotics and others) is expanding beyond traditional gastrointestinal targets, as current studies are investigating their potential benefits in various anatomical sites, including the oral cavity, vagina, and skin [187]. *Lactobacillaceae* family and *Bifidobacterium* appear to counteract the proliferation of *Yersinia*, *Lacticaseibacillus rhamnosus* L60 and *Limosilactobacillus fermentum* L23 that of *Candida albicans* and *Streptococcus lactis*, but also *Saccharomyces cerevisiae* instead against *Campylobacter jejuni* and *Staphylococcal* septicaemia. Furthermore, a significant reduction in the expression of *C. difficile* has been demonstrated with the probiotic strains *L. casei* and *B. breve* [188,189,190,191].

Fibre and resistant starch are not broken down by enzymes in the small intestine. They move to the large intestine where they undergo fermentation by resident microorganisms. Fibre is a good source of Microbiota-accessible carbohydrates (MACs) which can be used by microorganisms to provide the host with energy and carbon [192].

This property of plant fibres warrants their additional characterization as prebiotic foods, which are non-digestible nutrients that benefit the health of the host by selectively stimulating the growth and/or activity of certain microorganisms [193,194].

Sources of prebiotics include soy, inulins, raw wheat and barley, raw oats and non-digestible oligosaccharides, such as fructans, polydextrose, fructo-oligosaccharides, lacto-oligosaccharides, xylo-oligosaccharides and arabino-oligosaccharides [195].

A diet low in these substances reduces the total population of microorganisms. Conversely, a high intake of indigestible carbohydrates increases *Bifidobacteria* and *Lactic* acid bacteria [196].

Probiotic intake during infections has been noted to regulate the expression of certain taxa and, consequently, their healthy metabolites, have a benefit to the intestinal epithelium and are against dysbiosis. The link between fibres and infections would seem to be mainly based on the metabolome with the production of short-chain fatty acids (SCFAs) and their precursor succinic acid, which have demonstrated immunomodulatory activities at various levels [197].

Dietary polyphenols, which include catechins, flavanols, flavones, anthocyanins, proanthocyanidins, and phenolic acids, have been systematically studied for their antioxidant properties [198].

Common foods rich in polyphenols include fruits, seeds, vegetables, cocoa products, and wine. The most common bacteria associated with these food sources are *Bifidobacterium* and those from *Lactobacillaceae* family [199].

An increase in the population of *Bacteroides* has been observed in individuals consuming red wine polyphenols. Oral or systemic administration of polyphenols in animal experiments (rodents) has been noted to protect them from endotoxemia and sepsis since they show anti-inflammatory and vasculoprotective effects in clinical and preclinical studies [200,201].

Faecal microbiota transplant (FMT) is the process of transplanting faecal material from a healthy donor into a diseased recipient, with the aim of treating an underlying disease. Different definitions have been used in the past, such as faecal bacteriotherapy, faecal transfusion, or human probiotic infusion [202]. The aim is to restore the eubiosis of the IM so that metabolism, immunity and the ability to prevent colonization with pathogens will improve. FMT has been shown to be an effective treatment for refractory *C. difficile* infections (CDI) by competing with it, restoring secondary bile acids that inhibit *C. difficile* germination. Since the IM becomes dysbiosis during sepsis, this can lead to worsening its course and contribute to MODS [203,204]. Thus, many researchers through experiments in animals and septic patients have had satisfactory results on the management of dysbiosis and thus on the course of the disease itself. In many cases, repeated molecular tests have shown that the IM has been reconstructed with healthy bacteria [205].

Finally, targeted genomic sequencing is an emerging strategy to screen disease-specific microbiota biomarkers for clinical diagnosis and prognosis. However, this approach often produces inconsistent or conflicting results due to inadequate study design and sequencing bias [206]. Since microbiomes at various body sites differ and diseases do not occur in isolation, a comprehensive analysis strategy that highlights the full potential of microbiomes should include diverse sample types and various diseases, pan-body pan-disease microbiomics as a source for diagnostic and therapeutic strategies. The human microbiome emerges as a promising reservoir of diagnostic and therapeutic markers for theragnostic personalized treatment in medicine [206,207].

## 8. Conclusions

In recent scientific endeavours, research and studies have inspired and brought progress in revealing and clarifying the complex molecular and biochemical mechanisms between sepsis and IM. Modern knowledge consists in presenting novel and determinant interactions thus identifying the correlation between the fundamental pathogenetic pathways in clarifying and explaining complex functions and immune responses between IM and septic status. Thus, the pioneering relationship and exposure to the simultaneous activation and interaction of pro-inflammatory and anti-inflammatory factors, the proposal of the dynamic coexistence of an excessive and uncontrolled response in combination with an immunoparesis and immunosuppression response have contributed to highlighting and demonstrating the obscure and unclear complexity of the pathways of intestinal damage, dysbiosis and bacterial translocation that can cause septic status and vice versa. Thus, we have highlighted in this review:Sepsis induces IM dysbiosisStrengthening the altered intestinal barrier function (which can be highlighted with various methods) is a good tactic to reduce a possible translocation of bacteria and their metabolites and thus alleviate the MODS induced by sepsis or the risk of a septic statusIM itself participates in the development of sepsis and influences the host susceptibility to sepsis via the IM axes with gut-brain, intestine-liver etc.IM includes bacteria, fungi, viruses and archaea. The roles of different bacteria, fungi, viruses and archaea in the pathophysiology of sepsis need further investigation.Current limitations concern whether the investigation of the IM metabolome can be used as a biomarker for the development and outcome of sepsis.Administration of functional foods and faecal transplantation can help the course of a septic state or prevent it.

## Figures and Tables

**Figure 1 ijms-26-02028-f001:**
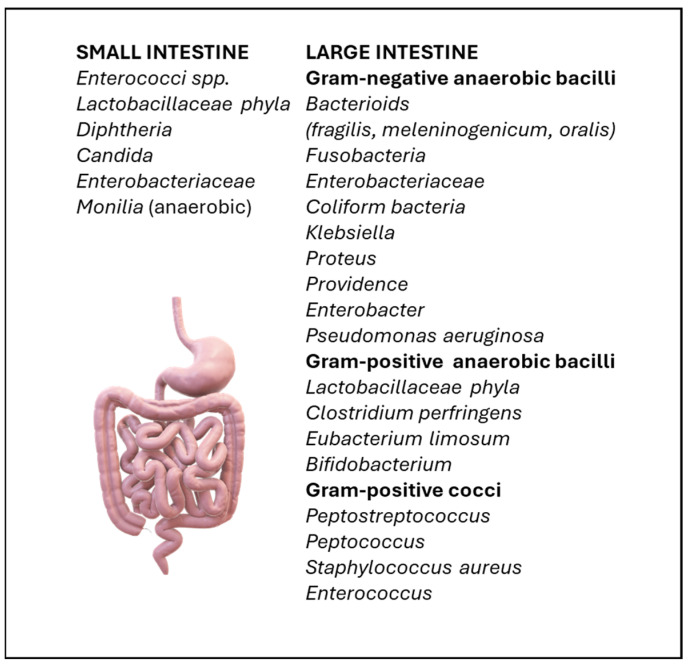
The IM contains a huge number of bacteria that coexist with humans, playing a key role in homeostasis and consists of aerobic and anaerobic microorganisms. Original figure by I.A. Charitos.

**Figure 2 ijms-26-02028-f002:**
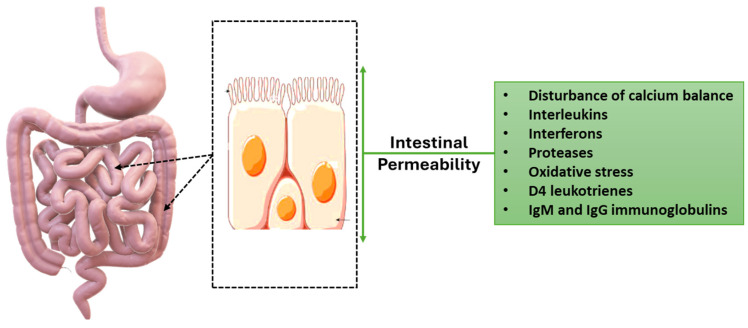
Several factors with regulatory action on the mucosal intestinal barrier through paracellular permeability. The green arrows indicate the direction routes of passage by the various molecules but also by the bacteria, that is, towards the inside or towards the outside that can be influenced by the factors described of the green table. The drawn arrows indicate the intestinal position of the affected mucosa. Original figure by I.A. Charitos.

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
