# Peer review of "Intestinal Microbiota Dysbiosis Role and Bacterial Translocation as a Factor for Septic Risk"

_ijms, 2025, doi:10.3390/ijms26052028_

Round 1

Reviewer 1 Report

Comments and Suggestions for Authors

The review file has been attached.

Author Response

A research article (ijms-3352845), entitled “On intestinal microbiotas’ role as a factor for septic risk“, submitted Antonella Cotoia et al., targets one of the most important topics in the gastroenterology and gastroimmunology nowadays.

The human immune system is deeply connected to the microbiota, which is first introduced to infants via maternal transfer at birth and evolves into a diverse microbial community influenced by diet and host immunity. This intestinal microbiota (IM) both shapes and is shaped by immune responses, with conditions such as infections, sepsis, and local intestinal inflammation significantly impacting microbial balance and promoting the proliferation or movement of harmful microorganisms. In general, the article is well-written (only minor English proofreading and text editing are required), well-structured, and clearly presents and discusses the outcomes of the study. However, the study exhibits a few of major drawbacks that should be addressed if it can be considered publishable in International Journal of Molecular Sciences.

  1. A graphical abstract is missing. Please, add it.

Answer

Thank you for the advice.

We use the ultimate figure 11 for that and we have modified it. All the changes are in yellow evidenced  (Please note that all the advice - changes are in yellow highlighted)

  1. A manuscript lacks graphics, there are only geometrical schemes with too much text in them – please replace these descriptions with images. a. English language is pleasing, however there are a few spelling errors.

Answer

 Thank you for the advice.

Most of the schemes are eliminated according to fourth reviewer

  1. List of abbreviations is missing. Please, summarize all acronyms and abbreviations in a table, alphabetiacally, in acronym-description manner, and present it before the Introduction.

Answer

We made one.

  1. Keywords should be ordered alphabetically.

Answer

We did it

  1. Authors didn’t describe the limitations of their study. Add it either in “Discussion” or in “Conclusions

Answer

We did it in the conclusions

  1. Authros nicely presented both dysregulated microbiota-afflicted immunity and molecular and pathogenic infections causing sepsis. Authors should also discuss toxin caused microbiota-gutbrain (MGB) dysregulation and dysbiosis that may lead to sepsis (https://doi.org/10.1016/j.ecoenv.2024.115965, please, cite it).

Answer

We did it.

  1. While discussing the dysbiosis, Authors should describe the microbial content and diversity as well as the Firmicutes-to-Bacteroidetes ratio in the healthy and diseased intestines.

Answer

We did it.

  1. Please, add a section about the link following: sepsis-enteric nervous systems neuroimmunomodulation-central nervous system (via the vagus nerve and MGB), which could a valuable novelty to the field.

Answer

We did it . 

  1. The review lacks a quantitative data. The most essential original studies cited should be presented and critically discussed in terms of precise intestinal toxins and sepsis inducers (lethal doses in humans, way of infection, IC50 or LD50 of the microorganism, etc.).

Answer

We did it . 

Authors should also be more specific about expetimental details employed in most crucial cited references.

Answer

We did it for some studies

A review is mostly descriptive and narrative, but it lack a constructive criticism being Authors’ opinion, which would cover originality and novelty to the field.

Answer

We did it. See conclusions

A chapter concnering therapeutic approaches must be added to the manuscript.

 Answer

We did it for functional foods.

The essential clinical trials (with pending ones) should be included in the review. Please, cover effectivess of pre-biotics, probiotics in restoring intestinal beneficial flora, fecal microbiota transplants, and dietary modifications.

Answer

We did it. Se fuctional foods and fecal transplantation

  1. Since microbiome Clinical translational of the crucial findings should be referred to the ongoing approaches of personalized (precision) medicine, precised theranostics, personalized metabolomics, and pan-body and pan-disease microbiomics (https://doi.org/10.1038/s41467-024- 52598-7). https://www.bmkgene.com/el/news/pan-genome-studies-provide-deep-and-full-genetic-views-of-a-species/

Answer

We did it. See in chapter for functional foods.

Reviewer 2 Report

Comments and Suggestions for Authors

The article show important issue - the context microbiota in sepsis risk. Actually, the role of micriobta is wide and is object of a lot of studies. The article is well structurized and has wide spectrum of problems. I recommend to accept this article. 

Author Response

Thank you for you comment. 

Reviewer 3 Report

Comments and Suggestions for Authors

I appreciate the Authors’ good intentions in writing the paper. However, this is a very difficult to read and understand manuscript, even for those specialized in the field (English mistakes left aside). Many terms are providing more confusion. On one hand, there are a lot of general data, not needed here. Any info not related to the topic should be removed. On the other hand, the review has no aim. Therefore, what are the readers looking for? Gut microbiota and sepsis? After reading the whole manuscript, I can say that “One cannot see the forest for the trees”. The way the manuscript appears now is like a bunch of data related to infection, sepsis (all stages) and microbiota in various conditions (not just sepsis), with many repetitions. I suggest the Authors to rewrite the whole manuscript, focus on their topic - altered microbiota, gut permeability etc and sepsis, with providing clear connections between them. Comments/suggestions:

1.       Tite: I like that the authors wanted to be less conventional; however, I would suggest finding a better and scientific one, not “On intestinal….”. Same for paragraphs in the main manuscript, starting with “On…”. Besides, intestinal microbiota PER SE, in healthy people does not represent a risk factor for sepsis. The Authors should consider that and formulate a proper title.

 2.       Abstract:

a.       I would suggest rewriting the sentence “Vertical transfer from the mother's microbiota to the infant at birth is considered the initial introduction of the intestinal microbiota (IM) to the infant.”, as there is scientific evidence that microbiota colonization in the gut may appear while in utero, during pregnancy.

b.       Please rewrite the sentence “Colonization resistance in the microbiota is influenced by local but also extra-intestinal factors such as in infections or sepsis.” – with the following in mind: what do you mean by “colonisation resistance in the microbiota”? and “infections or sepsis” – sepsis represents a severe form of infection.

c.        Please insert clearly the aim of your review.

d.       In all, the abstract should be rewritten, so that it states clearly what this manuscript is about. And, please correct all errors.

 3.       Introduction

I.  “On infection concepts”

a.       This subparagraph must be rewritten and precise terms defined correctly. It is not clear what is needed to be introduced. Please make sentences shorter and clarify content. Severe sepsis is introduced here, without defining sepsis, but the next paragraph refers to sepsis (2.1).

b.       Line 51: Please use “e.g.”, instead of “i.g.”  (i.g. is not correct; e.g. means “for example.” - “exempli gratia” in Latin.). Alternatively, depending on what you mean, you can use “i.e.” – meaning “that is”. (id est – Latin).

c.        Lines 51-52: please correct “mycelia”, in this instance.

d.       Lines 54-55: Please explain why latent is used twice in this instance: “By the term latent or subclinical or latent infection…”

e.       Lines 63-65: ”When the composition of the microbiota is altered (dysbiosi) and this alteration characterized by the presence of pathogenic microbes, then we say that there is pathological colonization.” Please correct the content and the term dysbiosis. I also suggest the Authors to read and use two important and recent manuscripts, in improving the quality of their paper and its inadvertences:

“Britton RA, Verdu EF, Di Rienzi SC, Reyes Muñoz A, Tarr PI, Preidis GA; Biotherapeutics Subcommittee of the AGA Center for Gut Microbiome Research and Education. Taking Microbiome Science to the Next Level: Recommendations to Advance the Emerging Field of Microbiome-Based Therapeutics and Diagnostics. Gastroenterology. 2024 Nov;167(6):1059-1064.”

“Porcari S, Mullish BH, Asnicar F, Ng SC, Zhao L, Hansen R, O'Toole PW, Raes J, Hold G, Putignani L, Hvas CL, Zeller G, Koren O, Tun H, Valles-Colomer M, Collado MC, Fischer M, Allegretti J, Iqbal T, Chassaing B, Keller J, Baunwall SM, Abreu M, Barbara G, Zhang F, Ponziani FR, Costello SP, Paramsothy S, Kao D, Kelly C, Kupcinskas J, Youngster I, Franceschi F, Khanna S, Vehreschild M, Link A, De Maio F, Pasolli E, Miguez AB, Brigidi P, Posteraro B, Scaldaferri F, Stojanovic MR, Megraud F, Malfertheiner P, Masucci L, Arumugam M, Kaakoush N, Segal E, Bajaj J, Leong R, Cryan J, Weersma RK, Knight R, Guarner F, Shanahan F, Cani PD, Elinav E, Sanguinetti M, de Vos WM, El-Omar E, Dorè J, Marchesi J, Tilg H, Sokol H, Segata N, Cammarota G, Gasbarrini A, Ianiro G. International consensus statement on microbiome testing in clinical practice. Lancet Gastroenterol Hepatol. 2024 Dec 5:S2468-1253(24)00311-X.”

f.         This paragraph, as it appears, is not clear and contains many confusing terms.

g.        Figure 1 is based on a reference from 2008 and content is not correct. Please revise.

h.       Figure 1 legend – this sentence should be part of the main text; it has no sense here. Please insert a proper figure legend, if you still want to keep this figure.

 II. “The body's defence system during an infection”

a.       Please provide data referring to intestinal defense system, and not body in general (same for Figure 2). Besides, all this info is well known.

b.       Figure 3 also contains known basic data (plus some mistakes). Please correct and adapt to your topic. The problem is that, since there is no aim of your review (usually mentioned by the end of the Introduction), the reader has no idea what will be the topic of the review.

 4.       Biochemical mechanisms during sepsis

I. “On the sepsis concept”

a. Please rewrite this paragraph, and define clearly important terms. And avoid redundancy. e.g There is again about SIRS, but it was already mentioned in Introduction: “Any infection can cause a systemic inflammatory reaction (SIRS). When it manifests itself with two or more of its points then we have systemic infections [12].” What does it mean?

b. Lines 151-153: sentence is just a repetition of the same terms in lines 141-143 – not needed; on the contrary, terms should be defined clearly: “septicemia, septic syndrome and septic shock, as mentioned, do not express distinct entities but a continuum, of increasing clinical severity, of the same basic disorder.”

c. Line 153: “The pathological process of sepsis is divided into five stages”. Beside the fact that sentences to follow contain some confusing terms, when it comes to the 5th stage, this is wrongly explained “Finally, in the last fifth stage, the dominant element is the organism's inability to control its reaction. It is the final stage of a now uncontrolled infection. This is a malignant development [17].”

d. Please correct “histokines”

e. Figure 4 contains also wrong terms. What is its role there?

f. This subparagraph should be nicely summarized and make all terms clearer. Too much unclear info, while it is not really needed.

II. “The pathogenesis clinical signs” (what does it mean?)

a.       Lines 236-237 – again, about SIRS.

b.       Lines 240-241 – “A complex immunological entity whereby SIRS and CARS interact throughout. In any case, the rot follows one direction.”. Please write proper sentences and reduce any words not needed.

c.        Line 247 “Putrefaction” ?

d.       Figure 5. Not that it is really needed, but if you decide to keep it, please define all the abbreviations.

e.       Leukocytosis, leukopenia, thrombopenia, hyperglycemia etc – are not clinical signs.

f.         Please place “figure 6” in the main text. Please correct all mistakes in Figure 6 (e.g. – leukocytosis > 12000/kg/thousand etc). Please define all abbreviations.

 5. The intestinal microbiota (IM) revelation

- Why is it a revelation in 2024 (beginning of 2025), in this paper? I did not find anything of that sort.

I. “Development of the community of microorganisms at the host”

a. Please correct all mistakes in figure 7. Plus, why “Fasting” in the “small intestine”?

b. “Intestinal colonization begins immediately after birth and is influenced by many factors.” Please correct, as per my above comment. It may appear in utero.

c. This paragraph is way too long and it contains many data, already known, plus with mistakes. Just one example: ”The large intestine teems with several microorganisms, and their constant proliferation is balanced by their constant excretion in the faeces. Most of them are not resistant to the acidic environment of the stomach, bile salts and gastrointestinal enzymes and thus are quickly inactivated [42].” What do bacteria from the colon have to do with the stomach? Plus, they are the most important.

II. “Microbiotas’ antimicrobial protection pathways”

a.       Again, it contains many sentences that are not related to the topic. Please present protection pathways against sepsis or for sepsis, as it is not clear from the title and no aim is presented for this review. Why to present respiratory system? Why neurological /psychiatric disorders?

b.       And why to talk here about Probiotics? This should be about gut microbiota, not “Live microorganisms that, when administered in adequate amounts, confer a health benefit on the host.” (ISAPP definition of Probiotics). All related sentences are not part of the topic, as probiotics do not belong to the intestinal microbiota. They are administered in adequate amounts...

c.        Also, these sentences contain a lot of mistakes – e.g. “"Akkermansia Propionippum" – What is this? It does not exist.

III. “The intestinal microbiotas’ dysbiosis role”

a.       Reference 63 (from 2015) is too old for it to be used in definition of “dysbiosis”. Same for reference [60], from 2016. We are in 2024! Hopefully 2025 when I finish this review.

b.       “There is increasing evidence that IM dysbiosis is associated with the pathogenesis of both intestinal and extra-intestinal disorders.” And sentences following – these aspects are known, please do refer to sepsis. And more sentences later referring to “Here we must mention the pneumogastric nerve or vagus nerve in the intestinal brain axis. In fact, it is mainly through vagal innervation that the brain regulates intestinal motility, visceral responses to pain, fear, anxiety and apprehension.”. So many sentences that do not belong to the topic.

 6.       Intestinal permeability pathways and bacteria translocation

This subparagraph is again full of info that does not refer to the topic. Beside the many sentences at the beginning, why talk about neoplasms here? “Decreased apoptosis correlates with the development of malignant neoplasms”. Please stick to sepsis.

I.                     “Factors that limit intestinal damage and intestinal bacterial translocation”

a.       This subparagraph also contains many unnecessary data, some wrong info and dangerous. Again – a repetition of “The large intestine teems with microbes, and their constant proliferation is balanced by their constant excretion in the faces. Most of them are not resistant to the acidic environment of the stomach, bile salts and gastrointestinal enzymes and thus are quickly inactivated”

b.       “However, the fetal intestinal tract is sterile and contains meconium. After the excretion of meconium and by various mechanisms, the development of the IM begins.” This is false, again.

 II.                   “Factors that damage mucosal intestinal barrier and promote bacterial translocation”

a.       Please present only info related to sepsis, both in main text and in figures. Why other diseases here? e.g. “Changes in their expression are related to the pathogenesis of both Crohn's disease and ulcerative colitis, which are characterized by chronicity and alternating periods of exacerbation and remission. During the period of disease exacerbation, neutrophilic polymorphonuclear cells migrate through the intestinal epithelial layer and cause mucosal inflammation” and the sentences following.

b.       Figure 9 is totally not needed here. It should be about sepsis, not all “pathological conditions of the gastrointestinal system and other systems of the body”, including malignancies, multiple sclerosis, asthma etc etc.

 7.       The intestinal microbiotas’ bacterial translocation role in sepsis

Figure 10 – here – showing factors that are involved in the intestinal barrier – is not related to the topic and, in any case, it was addressed previously.

8.       The Assessment of intestinal permeability

I do not see why this subparagraph is inserted here. Sentences should be correct, anyway.

Before Conclusion, I affirm with all due respect, that I did not find anything strongly proving that indigenous intestinal microbiota is a risk factor for sepsis.

9.       Conclusion

a.       Please rewrite this sentence, in order to be understood: “The risk of infections that can evolve into sepsis caused by the same IM due to dysbiosis that favors the entry and translocation of pathogenic or potentially pathogenic bacteria o fungi is not a wrong hypothesis.”. Beside the English language corrections required, please write clear sentences. Why should be a “not wrong hypothesis”? And what “hypothesis”? As nothing was explained clearly.

b.       What do you mean by this “Thus, correlate intestinal barrier alterations with the pathogenesis of several bacteria (such as the Gram- negative) that lead to sepsis, as well as demonstrating that these early alterations are also present in other critically ill patients such as polytraumatized patients and severely burned patients.”?

c.        Nothing in Conclusion is clear. And why about functional foods here?

d.       Why inserting a figure here? Plus, containing many errors (including figure legend – “amphidromic” and the rest?).

e.       What are the highlights of this review? In what way is it useful? Please revise the whole manuscript, focus on your topic and write proper conclusions. And elaborate appropriate directions for future research.

Comments on the Quality of English Language

Major revision is required, including for typos, misspelled words, missing words, comma use, verb-noun agreement, syntax, overall style. It is important to also avoid redundant info and long sentences.

Author Response

I appreciate the Authors’ good intentions in writing the paper. However, this is a very difficult to read and understand manuscript, even for those specialized in the field (English mistakes left aside). Many terms are providing more confusion. On one hand, there are a lot of general data, not needed here. Any info not related to the topic should be removed. On the other hand, the review has no aim. Therefore, what are the readers looking for? Gut microbiota and sepsis? After reading the whole manuscript, I can say that “One cannot see the forest for the trees”. The way the manuscript appears now is like a bunch of data related to infection, sepsis (all stages) and microbiota in various conditions (not just sepsis), with many repetitions. I suggest the Authors to rewrite the whole manuscript, focus on their topic - altered microbiota, gut permeability etc and sepsis, with providing clear connections between them. Comments/suggestions:

  1. Tite: I like that the authors wanted to be less conventional; however, I would suggest finding a better and scientific one, not “On intestinal….”. Same for paragraphs in the main manuscript, starting with “On…”. Besides, intestinal microbiota PER SE, in healthy people does not represent a risk factor for sepsis. The Authors should consider that and formulate a proper title.

Answer

We did it (Please note that all the advice - changes are in purple highlighted)

  1. Abstract:
  2. I would suggest rewriting the sentence “Vertical transfer from the mother's microbiota to the infant at birth is considered the initial introduction of the intestinal microbiota (IM) to the infant.”, as there is scientific evidence that microbiota colonization in the gut may appear while in utero, during pregnancy.

Answer

    We did it.

  1. Please rewrite the sentence “Colonization resistance in the microbiota is influenced by local but also extra-intestinal factors such as in infections or sepsis.” – with the following in mind: what do you mean by “colonisation resistance in the microbiota”? and “infections or sepsis” – sepsis represents a severe form of infection.

Answer

 We change it.

  1. Please insert clearly the aim of your review.

Answer

 We did it.

  1. In all, the abstract should be rewritten, so that it states clearly what this manuscript is about. And, please correct all errors.

Answer

We did it.

  1. Introduction
  2. “On infection concepts”
  3. This subparagraph must be rewritten and precise terms defined correctly. It is not clear what is needed to be introduced. Please make sentences shorter and clarify content. Severe sepsis is introduced here, without defining sepsis, but the next paragraph refers to sepsis (2.1).
  4. Line 51: Please use “e.g.”, instead of “i.g.”  (i.g. is not correct; e.g. means “for example.” - “exempli gratia” in Latin.). Alternatively, depending on what you mean, you can use “i.e.” – meaning “that is”. (id est – Latin).

Answer

 We did it. Introduction is all modificate

  1. Lines 54-55: Please explain why latent is used twice in this instance: “By the term latent or subclinical or latent infection…”

Answer

We did it. Introduction have several modifications

  1. Lines 63-65: ”When the composition of the microbiota is altered (dysbiosi) and this alteration characterized by the presence of pathogenic microbes, then we say that there is pathological colonization.” Please correct the content and the term dysbiosis. I also suggest the Authors to read and use two important and recent manuscripts, in improving the quality of their paper and its inadvertences:

“Britton RA, Verdu EF, Di Rienzi SC, Reyes Muñoz A, Tarr PI, Preidis GA; Biotherapeutics Subcommittee of the AGA Center for Gut Microbiome Research and Education. Taking Microbiome Science to the Next Level: Recommendations to Advance the Emerging Field of Microbiome-Based Therapeutics and Diagnostics. Gastroenterology. 2024 Nov;167(6):1059-1064.”

“Porcari S, Mullish BH, Asnicar F, Ng SC, Zhao L, Hansen R, O'Toole PW, Raes J, Hold G, Putignani L, Hvas CL, Zeller G, Koren O, Tun H, Valles-Colomer M, Collado MC, Fischer M, Allegretti J, Iqbal T, Chassaing B, Keller J, Baunwall SM, Abreu M, Barbara G, Zhang F, Ponziani FR, Costello SP, Paramsothy S, Kao D, Kelly C, Kupcinskas J, Youngster I, Franceschi F, Khanna S, Vehreschild M, Link A, De Maio F, Pasolli E, Miguez AB, Brigidi P, Posteraro B, Scaldaferri F, Stojanovic MR, Megraud F, Malfertheiner P, Masucci L, Arumugam M, Kaakoush N, Segal E, Bajaj J, Leong R, Cryan J, Weersma RK, Knight R, Guarner F, Shanahan F, Cani PD, Elinav E, Sanguinetti M, de Vos WM, El-Omar E, Dorè J, Marchesi J, Tilg H, Sokol H, Segata N, Cammarota G, Gasbarrini A, Ianiro G. International consensus statement on microbiome testing in clinical practice. Lancet Gastroenterol Hepatol. 2024 Dec 5:S2468-1253(24)00311-X.”

Answer

We did it. Introduction have several modifications

  1. This paragraph, as it appears, is not clear and contains many confusing terms.

Answer

We did it. Introduction have several modifications

  1. Figure 1 is based on a reference from 2008 and content is not correct. Please revise.

Answer

We did it. Introduction have several modifications. Cancel fig.1

  1. Figure 1 legend – this sentence should be part of the main text; it has no sense here. Please insert a proper figure legend, if you still want to keep this figure.

Answer

We did it. Introduction have several modifications. Cancel fig.1

  1. “The body's defence system during an infection”
  2. Please provide data referring to intestinal defense system, and not body in general (same for Figure 2). Besides, all this info is well known.

Answer

We did it. Introduction have several modifications. Cancel fig.2

  1. Figure 3 also contains known basic data (plus some mistakes). Please correct and adapt to your topic. The problem is that, since there is no aim of your review (usually mentioned by the end of the Introduction), the reader has no idea what will be the topic of the review.

Answer

We did it. Introduction have several modifications. Cancel fig.3

  1. Biochemical mechanisms during sepsis
  2. “On the sepsis concept”
  3. Please rewrite this paragraph, and define clearly important terms. And avoid redundancy. e.g There is again about SIRS, but it was already mentioned in Introduction: “Any infection can cause a systemic inflammatory reaction (SIRS). When it manifests itself with two or more of its points then we have systemic infections [12].” What does it mean?

Answer

We did it. Introduction have several modifications.

  1. Lines 151-153: sentence is just a repetition of the same terms in lines 141-143 – not needed; on the contrary, terms should be defined clearly: “septicemia, septic syndrome and septic shock, as mentioned, do not express distinct entities but a continuum, of increasing clinical severity, of the same basic disorder.”

Answer

We did it. Introduction have several modifications

  1. Line 153: “The pathological process of sepsis is divided into five stages”. Beside the fact that sentences to follow contain some confusing terms, when it comes to the 5thstage, this is wrongly explained “Finally, in the last fifth stage, the dominant element is the organism's inability to control its reaction. It is the final stage of a now uncontrolled infection. This is a malignant development [17].

Answer

We did it. Introduction have several modifications

  1. Please correct “histokines”

Answer

We did it. Introduction have several modifications. Cancel tex.

  1. Figure 4 contains also wrong terms. What is its role there?

Answer

We did it. Introduction have several modifications. Cancel text and figure 4.

  1. This subparagraph should be nicely summarized and make all terms clearer. Too much unclear info, while it is not really needed.

Answer

We did it. Introduction have several modifications. Cancel text.

  1. “The pathogenesis clinical signs” (what does it mean?)
  2. Lines 236-237 – again, about SIRS.

Answer

We did it. Introduction have several modifications.

  1. Lines 240-241 – “A complex immunological entity whereby SIRS and CARS interact throughout. In any case, the rot follows one direction.”. Please write proper sentences and reduce any words not needed.

Answer

We did it. Introduction have several modifications. Cancel text

  1. Line 247 “Putrefaction” ?

Answer

We did it. Introduction have several modifications. Cancel text

  1. Figure 5. Not that it is really needed, but if you decide to keep it, please define all the abbreviations.

Answer

We did it. Introduction have several modifications. Cancel text and fig.5

  1. Leukocytosis, leukopenia, thrombopenia, hyperglycemia etc – are not clinical signs.

Answer

We did it. Introduction have several modifications. Cancel text.

  1. Please place “figure 6” in the main text. Please correct all mistakes in Figure 6 (e.g. – leukocytosis > 12000/kg/thousand etc). Please define all abbreviations.

Answer

We define all abbreviations. We think that is better as table 1.

  1. The intestinal microbiota (IM) revelation

- Why is it a revelation in 2024 (beginning of 2025), in this paper? I did not find anything of that sort.

Answer

We define better.

  1. “Development of the community of microorganisms at the host”
  2. Please correct all mistakes in figure 7. Plus, why “Fasting” in the “small intestine”?

Answer

We change the text

  1. “Intestinal colonization begins immediately after birth and is influenced by many factors.” Please correct, as per my above comment. It may appear in utero.

Answer

We change the text

  1. This paragraph is way too long and it contains many data, already known, plus with mistakes. Just one example: ”The large intestine teems with several microorganisms, and their constant proliferation is balanced by their constant excretion in the faeces. Most of them are not resistant to the acidic environment of the stomach, bile salts and gastrointestinal enzymes and thus are quickly inactivated [42].” What do bacteria from the colon have to do with the stomach? Plus, they are the most important.

Answer

We change the text

  1. “Microbiotas’ antimicrobial protection pathways”
  2. Again, it contains many sentences that are not related to the topic. Please present protection pathways against sepsis or for sepsis, as it is not clear from the title and no aim is presented for this review. Why to present respiratory system? Why neurological /psychiatric disorders?
  3. Answer

We change the text.

  1. And why to talk here about Probiotics? This should be about gut microbiota, not “Live microorganisms that, when administered in adequate amounts, confer a health benefit on the host.” (ISAPP definition of Probiotics). All related sentences are not part of the topic, as probiotics do not belong to the intestinal microbiota. They are administered in adequate amounts...
  2. Answer

We change the text and make a chapter about functional foods.

  1. Also, these sentences contain a lot of mistakes – e.g. “"Akkermansia Propionippum" – What is this? It does not exist.

Answer

We change in the text

III. “The intestinal microbiotas’ dysbiosis role”

  1. Reference 63 (from 2015) is too old for it to be used in definition of “dysbiosis”. Same for reference [60], from 2016. We are in 2024! Hopefully 2025 when I finish this review.

Answer

We ad in the text 2 recent.

  1. “There is increasing evidence that IM dysbiosis is associated with the pathogenesis of both intestinal and extra-intestinal disorders.” And sentences following – these aspects are known, please do refer to sepsis. And more sentences later referring to “Here we must mention the pneumogastric nerve or vagus nerve in the intestinal brain axis. In fact, it is mainly through vagal innervation that the brain regulates intestinal motility, visceral responses to pain, fear, anxiety and apprehension.”. So many sentences that do not belong to the topic.

Answer

We change in the text

  1. Intestinal permeability pathways and bacteria translocation

This subparagraph is again full of info that does not refer to the topic. Beside the many sentences at the beginning, why talk about neoplasms here? “Decreased apoptosis correlates with the development of malignant neoplasms”. Please stick to sepsis.

Answer

We cancel in the text

  1. “Factors that limit intestinal damage and intestinal bacterial translocation”
  2. This subparagraph also contains many unnecessary data, some wrong info and dangerous. Again – a repetition of “The large intestine teems with microbes, and their constant proliferation is balanced by their constant excretion in the faces. Most of them are not resistant to the acidic environment of the stomach, bile salts and gastrointestinal enzymes and thus are quickly inactivated”

Answer

We change in the text

  1. “However, the fetal intestinal tract is sterile and contains meconium. After the excretion of meconium and by various mechanisms, the development of the IM begins.” This is false, again.

Answer

We change in the text

  1. “Factors that damage mucosal intestinal barrier and promote bacterial translocation”
  2. Please present only info related to sepsis, both in main text and in figures. Why other diseases here? e.g. “Changes in their expression are related to the pathogenesis of both Crohn's disease and ulcerative colitis, which are characterized by chronicity and alternating periods of exacerbation and remission. During the period of disease exacerbation, neutrophilic polymorphonuclear cells migrate through the intestinal epithelial layer and cause mucosal inflammation” and the sentences following.

Answer

We change in the text

  1. Figure 9 is totally not needed here. It should be about sepsis, not all “pathological conditions of the gastrointestinal system and other systems of the body”, including malignancies, multiple sclerosis, asthma etc etc.
  2. Answer

We cancel in the text

  1. The intestinal microbiotas’ bacterial translocation role in sepsis

Figure 10 – here – showing factors that are involved in the intestinal barrier – is not related to the topic and, in any case, it was addressed previously.

Answer

We cancel in the text

  1. The Assessment of intestinal permeability

I do not see why this subparagraph is inserted here. Sentences should be correct, anyway.

Answer

New title:

Assessment of intestinal permeability as a strategy to strengthen intestinal barrier function to avoid bacterial translocation

Before Conclusion, I affirm with all due respect, that I did not find anything strongly proving that indigenous intestinal microbiota is a risk factor for sepsis.

Answer

New references and text added for this topic.

  1. Conclusion
  2. Please rewrite this sentence, in order to be understood: “The risk of infections that can evolve into sepsis caused by the same IM due to dysbiosis that favors the entry and translocation of pathogenic or potentially pathogenic bacteria o fungi is not a wrong hypothesis.”. Beside the English language corrections required, please write clear sentences. Why should be a “not wrong hypothesis”? And what “hypothesis”? As nothing was explained clearly.

Answer

New text and cancelled text added for this.

  1. What do you mean by this “Thus, correlate intestinal barrier alterations with the pathogenesis of several bacteria (such as the Gram- negative) that lead to sepsis, as well as demonstrating that these early alterations are also present in other critically ill patients such as polytraumatized patients and severely burned patients.”?

Answer

New text and cancelled text added for this

  1. Nothing in Conclusion is clear. And why about functional foods here?

Answer

New text and cancelled text added for this. We added new chapter for this topic.

  1. Why inserting a figure here? Plus, containing many errors (including figure legend – “amphidromic” and the rest?).

Answer

The figure now is as Graphical Abstract

  1. What are the highlights of this review? In what way is it useful? Please revise the whole manuscript, focus on your topic and write proper conclusions. And elaborate appropriate directions for future research.

Answer

New text added for this.

 Comments on the Quality of English Language: Major revision is required, including for typos, misspelled words, missing words, comma use, verb-noun agreement, syntax, overall style. It is important to also avoid redundant info and long sentences.

Answer

We did it.

Reviewer 4 Report

Comments and Suggestions for Authors

This review systematically describes the relationship between gut microbiota, pathogens, and sepsis, which is of significant importance. I believe that certain modifications are needed before it can be accepted by IJMS:

  1. In line 62, “Human” should be changed to “human”.
  2. In line 63, “dysbiosi” should be modified to “dysbiosis”.
  3. In lines 79-80, some words have different font formatting compared to others.
  4. In line 158, the authors should add appropriate references. Additionally, the inflammatory factors mentioned here are not exhaustive; they are only highlighted due to their significant expression in the early stages of infection.
  5. In lines 175-176, the authors mention that LPS is released from dead bacteria. Is this statement accurate?
  6. In line 188, what is “T17”? Is this a spelling error? Furthermore, the spelling of TH- is not accurate; it should be Th.
  7. In section 2.1, the authors mention five steps of pathogen invasion during sepsis. However, since this review discusses the role of gut microbiota, the authors should address the changes in gut microbiota during sepsis, particularly the relative abundance of probiotics and the abundance changes of microbiota related to immune metabolism, which are very important.
  8. Section 2.2 has too many paragraphs. The concept of a review is to summarize advanced research rather than simply stating facts. Please consolidate related logical paragraphs; similar issues appear in other sections as well.
  9. Please revise the subtitle of section 3.1. The phrase “at host” is not commonly used.
  10. In Fig. 7, the authors classify the infection of Gram-negative and Gram-positive bacteria in the small intestine. Such a classification does not exist for the microbiota in the small intestine. Additionally, please ensure that all bacterial scientific names are italicized.
  11. In lines 327-329, this sentence lacks relevant literature citations for support.
  12. In line 333, the authors mention the functions of anaerobic and aerobic bacteria; these should be discussed separately.
  13. In line 415, the authors should distinguish between diversity and richness.
  14. Similarly, in lines 436-441, the authors still do not provide corresponding literature support.
  15. In line 494, a reference (DeGruttola et al., 2016) appears suddenly. Is this a citation?
  16. This manuscript contains many abbreviations; it is recommended that the authors create a separate section to display an abbreviation list.
  17. This manuscript has numerous grammatical and spelling errors; the authors should revise carefully and improve the overall language quality.
  18. Please revise the figure legend for figure 1. The figure legend should simply state the content of the image and summarize it, rather than presenting a conclusion.
  19. The authors have not provided sufficient references to support many of their concepts throughout the manuscript, which requires thorough and patient revisions for a review.
  20. The word count in the conclusion section is too high; a conclusion should be limited to 150 words and should avoid references to images and tables in the main text

Author Response

This review systematically describes the relationship between gut microbiota, pathogens, and sepsis, which is of significant importance. I believe that certain modifications are needed before it can be accepted by IJMS:

  1. In line 62, “Human” should be changed to “human”.

Answer

We did it.  (Please note that all the advice - changes are in green highlighted)

  1. In line 63, “dysbiosi” should be modified to “dysbiosis”.

Answer

We did it.

  1. In lines 79-80, some words have different font formatting compared to others.

Answer

Text cancelled according to fourth reviewer

  1. In line 158, the authors should add appropriate references. Additionally, the inflammatory factors mentioned here are not exhaustive; they are only highlighted due to their significant expression in the early stages of infection.

Answer

Text cancelled according to fourth reviewer

  1. In lines 175-176, the authors mention that LPS is released from dead bacteria. Is this statement accurate?

Answer

Text cancelled according to fourth reviewer

  1. In line 188, what is “T17”? Is this a spelling error? Furthermore, the spelling of TH- is not accurate; it should be Th.

Answer

Text cancelled according to fourth reviewer

In section 2.1, the authors mention five steps of pathogen invasion during sepsis. However, since this review discusses the role of gut microbiota, the authors should address the changes in gut microbiota during sepsis, particularly the relative abundance of probiotics and the abundance changes of microbiota related to immune metabolism, which are very important.

Answer

Text cancelled according to fourth reviewer

  1. Section 2.2 has too many paragraphs. The concept of a review is to summarize advanced research rather than simply stating facts. Please consolidate related logical paragraphs; similar issues appear in other sections as well.

Answer

Text cancelled according to fourth reviewer

  1. Please revise the subtitle of section 3.1. The phrase “at host” is not commonly used.

Answer

We cancelled

  1. In Fig. 7, the authors classify the infection of Gram-negative and Gram-positive bacteria in the small intestine. Such a classification does not exist for the microbiota in the small intestine. Additionally, please ensure that all bacterial scientific names are italicized.

Answer

The text has been modified and moved for better understanding. It refers to the bacteria present in the normal non-pathological microbiota. Names are italicized

  1. In lines 327-329, this sentence lacks relevant literature citations for support.

Answer

We add the crucial references

  1. In line 333, the authors mention the functions of anaerobic and aerobic bacteria; these should be discussed separately.

Answer

Text cancelled

  1. In line 415, the authors should distinguish between diversity and richness.

Answer

Text cancelled according to fourth reviewer

  1. Similarly, in lines 436-441, the authors still do not provide corresponding literature support.

Answer

Text cancelled according to fourth reviewer

  1. In line 494, a reference (DeGruttola et al., 2016) appears suddenly. Is this a citation?

Answer

Yes, it is a citation and cancelled from the text.

  1. This manuscript contains many abbreviations; it is recommended that the authors create a separate section to display an abbreviation list.

Answer

we did it

  1. This manuscript has numerous grammatical and spelling errors; the authors should revise carefully and improve the overall language quality.

Answer

We did it

  1. Please revise the figure legend for figure 1. The figure legend should simply state the content of the image and summarize it, rather than presenting a conclusion.

Answer

We cancelled according to fourth reviewer

  1. The authors have not provided sufficient references to support many of their concepts throughout the manuscript, which requires thorough and patient revisions for a review.

Answer

We did it

  1. The word count in the conclusion section is too high; a conclusion should be limited to 150 words and should avoid references to images and tables in the main text

           Answer

We did it

Reviewer 5 Report

Comments and Suggestions for Authors

This paper in Int.J.Mol.Sci claims to look at the role of intestinal microbes as a factor for septic risk. It does not.

It also is not really a review. Nowhere in the paper do the authors formally state what they are reviewing, how they will do it, and why such a review is necessary. I recommend the authors go through the free resources at https://www.litr-ex.com/ , which explains how a literature review works and the different types of reviews. 

Overall, I do not think this paper can be salvaged. If the authors are serious about writing a literature review, they will need to do a lot of hard work, follow the guidelines for writing a review, and rewrite the paper entirely from the beginning.

The first few paragraphs of the introduction define terms [often unclearly, and in some cases incorrectly], but it is not clear why or whether anyone reading this paper would need to have them defined. Do you really think a person reading a paper on sepsis does not know what "infection" means? Many of the words defined here also do not appear later in the manuscript, or only appear once, and so defining them is pointless. Section 1.1 must be deleted completely.

Section 1.2 and figures 2 and 3 look at the skin and mucous membranes and immune system. The paper is about intestinal microbes. Section 1.2 must be deleted.

Section 2.1 defines sepsis [poorly] and then goes into a lot of detail about the immune response, but does not introduce anything about intestinal microbes or why then scientific community needs to look at the role of intestinal microbes as a factor for risk of sepsis. Section 2.2 looks at the symptoms of sepsis. All these sections need to be converted into one paragraph [no more!] that defines sepsis in as few words as possible, without mentioning anything that will not be mentioned later in the paper. This is the introduction to the review, so introduce the review, and nothing else.

Section 3.1 is the first section that is actually related to the topic of the paper. It makes many claims but cites very few sources: often one source per paragraph, and many of those sources are themselves review papers [which begs the question of why this paper is needed if better reviews already exist].

3.2 starts with introducing the IM, again. Introduce it once in the introduction, then do not re-introduce it again. It then discusses the "built environment," seeming to forget that this paper is about IM and sepsis. The word "sepsis" does not appear at all in section 3.2. I recommend it be deleted completely.

Section 3.3 also is not about sepsis and must be deleted completely.

Section 4 is also not about sepsis and must be deleted completely.

Section 5 is the first part of the paper that resembles what the title promised.

Section 6 barely mentions sepsis.

Ultimately, the relevant sentences of the manuscript, if combined, would be no more than four paragraphs. That's not a paper. This manuscript cannot be salvaged. The authors should research how to write a review paper, and start again, clearly defining their research goals.

Other comments
51 It is absolutely not correct to define mycelia as the presence of fungi in the blood. Please consult a dictionary before defining words.
19+52 You defined infection twice, and differently
70-71 One of the following what?
102 It is not appropriate to use "therefore" like this.
146 what points?
141+151 Another example of repetition.
314-315 this sentence seems random. Why is it here? What about the microbes in the rest of the gut? Every paragraph in any text must have one point, and every sentence in that paragraph must be related to that point.
795 more plausible explanation than what, and for what?
886-888 There is no hypothesis in this sentence.
888-889 This is common sense. I feel like many people, even non-scientists, know that a disrupted immune system and unhealthy factors lead to higher risk of infection. Quackery peddlers regularly make this claim. Above a certain age, many children know it. A review paper is not needed to claim something this basic.
906-909 An extraordinary claim requiring extraordinary evidence, none of which exists in this paper. If you want to claim something like this, then write a [proper!] review paper solely on that subject and nothing else [although dozens of review papers already exist on that subject, so another review is not necessary].

Comments on the Quality of English Language

The text is poorly written with a lot of typos, but the more pressing concern is it's not well structured. It's simply not a review paper.

Many paragraphs exist to summarize a finding from one paper. That is, once citation per paragraph. That's not a good way to write a paper.

Author Response

This paper in Int.J.Mol.Sci claims to look at the role of intestinal microbes as a factor for septic risk. It does not. It also is not really a review. Nowhere in the paper do the authors formally state what they are reviewing, how they will do it, and why such a review is necessary. I recommend the authors go through the free resources at https://www.litr-ex.com/ , which explains how a literature review works and the different types of reviews.  Overall, I do not think this paper can be salvaged. If the authors are serious about writing a literature review, they will need to do a lot of hard work, follow the guidelines for writing a review, and rewrite the paper entirely from the beginning.
Answer

Is a narrative review (see in text the  Introduction)
The first few paragraphs of the introduction define terms [often unclearly, and in some cases incorrectly], but it is not clear why or whether anyone reading this paper would need to have them defined. Do you really think a person reading a paper on sepsis does not know what "infection" means? Many of the words defined here also do not appear later in the manuscript, or only appear once, and so defining them is pointless.

  1. Section 1.1 must be deleted completely.

Answer

We did it. (Please note that all the advice - changes are in light blue highlighted)

Section 1.2 and figures 2 and 3 look at the skin and mucous membranes and immune system. The paper is about intestinal microbes. Section 1.2 must be deleted.

Answer

We did it

Section 2.1 defines sepsis [poorly] and then goes into a lot of detail about the immune response butdoes not introduce anything about intestinal microbes or why then scientific community needs to look at the role of intestinal microbes as a factor for risk of sepsis.

Answer

We did it

Section 2.2 looks at the symptoms of sepsis. All these sections need to be converted into one paragraph [no more!] that defines sepsis in as few words as possible, without mentioning anything that will not be mentioned later in the paper. This is the introduction to the review, so introduce the review, and nothing else.
Answer

We did it

Section 3.1 is the first section that is actually related to the topic of the paper. It makes many claims but cites very few sources: often one source per paragraph, and many of those sources are themselves review papers [which begs the question of why this paper is needed if better reviews already exist].
Answer

We add text and references according to the first reviewer

3.2 starts with introducing the IM, again. Introduce it once in the introduction, then do not re-introduce it again. It then discusses the "built environment," seeming to forget that this paper is about IM and sepsis. The word "sepsis" does not appear at all in section 3.2. I recommend it be deleted completely.

Answer

We did it
Section 3.3 also is not about sepsis and must be deleted completely.
Answer

We add text and references according to the first and third reviewer

Answer
Section 4 is also not about sepsis and must be deleted completely.
Answer

We add text and references according to the first reviewer

Section 5 is the first part of the paper that resembles what the title promised.
Section 6 barely mentions sepsis.

  1. Ultimately, the relevant sentences of the manuscript, if combined, would be no more than four paragraphs. That's not a paper. This manuscript cannot be salvaged. The authors should research how to write a review paper, and start again, clearly defining their research goals.
    Other comments

  2. 51 It is absolutely not correct to define mycelia as the presence of fungi in the blood. Please consult a dictionary before defining words.

Answer

We  cancelled 1.1.

19+52 You defined infection twice, and differently
Answer

We cancelled

70-71 One of the following what?
     Answer

     We cancelled

102 It is not appropriate to use "therefore" like this.

Answer

We cancelled

146 what points?
Answer

We cancelled

141+151 Another example of repetition.
Answer

We cancelled

  1. 314-315 this sentence seems random. Why is it here? What about the microbes in the rest of the gut?

Answer

We cancelled

Every paragraph in any text must have one point, and every sentence in that paragraph must be related to that point.
Answer

We did it

  1. 795 more plausible explanation than what, and for what?

Answer
We add: Thus, MODS is one of the causes of bacterial translocation.

886-888 There is no hypothesis in this sentence.
Answer
We have reviewed everything

  1. 888-889 This is common sense. I feel like many people, even non-scientists, know that a disrupted immune system and unhealthy factors lead to higher risk of infection. Quackery peddlers regularly make this claim. Above a certain age, many children know it. A review paper is not needed to claim something this basic.

Answer
We have reviewed everything

  1. 906-909 An extraordinary claim requiring extraordinary evidence, none of which exists in this paper. If you want to claim something like this, then write a [proper!] review paper solely on that subject and nothing else [although dozens of review papers already exist on that subject, so another review is not necessary].
  2. Answer
    We have reviewed everything

  1. Comments on the Quality of English Language
  2. The text is poorly written with a lot of typos, but the more pressing concern is it's not well structured. It's simply not a review paper.

Answer

We did it

  1. Many paragraphs exist to summarize a finding from one paper. That is, once citation per paragraph. That's not a good way to write a paper.

Answer

We did more than one citation.

Round 2

Reviewer 3 Report

Comments and Suggestions for Authors

The manuscript was markedly improved, with many corrections, according to the Reviewers’ suggestions / comments. Now, it makes more sense and adds appropriate organized data, without mistakes.  Both submitted versions - pdf and Supplementary file contain the revised manuscript with a lot of deleted and added text (Track Changes) and many colors. This is very good, but I did not see any clean copy, so that I can appreciate in detail how the final manuscript would look. I suggest the Authors to upload also a clean copy, which could be properly verified.

Comments on the Quality of English Language

With so many deleted sentences and added ones, it is impossible to verify all the comas etc, as they are not seen. Hopefully, the clean copy will allow a better appreciation.

Author Response

Dear reviewer, we thank you for the exceptional advice that you have given us. We upload a clean copy with only the second-round modifications for the fifth reviewer. 

Reviewer 4 Report

Comments and Suggestions for Authors

I have no more questions on this manuscript.

Author Response

Dear reviewer, we are glad that the manuscript has been improved thanks to your valuable comment.

Reviewer 5 Report

Comments and Suggestions for Authors

The authors made the changes requested, and added text. The new sections are great additions.

The new paper should be about the role of intestinal microbiota dysbiosis and bacterial translocation as factors for sepsis, according to the title. However, the authors still have not clearly defined what their review is actually about in the manuscript itself, and the content of the paper is broad and scattered, even though it's improved.

The text lacks cohesiveness: many paragraphs do not connect to each other, as if multiple authors all wrote different paragraphs separately, in different styles, that were then arranged into a paper without enough effort to make them fit together. Some of the text reads like it was taken from a different paper on a different topic. There are some paragraphs that are overly detailed, while many other paragraphs are one sentence long.  There is no clear vision on what the paper is supposed to be, or what each section's purpose is.

One solution, effective though time-consuming, is to delete everything and start over, but have a single author rewrite the entire manuscript from beginning to end. No more single-sentence paragraphs! No more redundancy. One person with a clear vision for what the paper should look like should write the manuscript from beginning to end. The same amount of co-authors is fine, but only one person should actually write the manuscript. Whoever wrote the long paragraphs did well. I recommend starting from a simple outline, with a few words for each section. Then, fill in the outline: describe each paragraph in a few words. Once you have this skeleton of the paper, add information and citations. This should ensure you do not repeat yourself. Start with the basic outline of the entire paper, and then build it into a full paper. Use big paragraphs, never small ones, and ensure that 1) every sentence in the paragraph is related to the topic introduced at the first sentence, and every sentence is related to the sentence that comes before and after. 2) Every sentence connecting to that topic is in that paragraph only, and nowhere else in the paper [A mistake made when not using an outline]. 3) Every paragraph in a section is strongly and obviously related to the topic of that section, no exceptions, and every paragraph is connected to the paragraph that comes before or after in the section. 4) You never say the same thing twice, anywhere in the paper. 5) Any information the reader does not need is deleted.

The new paper still makes a few claims with no references, though much less than before. It's improved, but not finished.

Some topics are limited to a sentence or two, when they themselves could be the subject of a full review paper. This review paper does not review. The authors should cite more than one example from the primary literature [not from other review papers] for any general statement.

Line numbers below are from the revised text pdf file. Sorry for the inconvenience.
134-140 Split this into two sentences.
153 It's not research, it's a review. And it's not about biomechancics, it's about bacterial translocation, no? State your topic very clearly: it should be as narrow as possible. A review about "biomechanics," for example, would need at least 1000 citations. Also, if you are going to have a review to gain a picture about what is not yet known about a topic, then be sure that you read almost every paper about that topic, and cite zero papers unrelated to that topic.
156-161 I think you are trying to say here that a non-narrative review (like a systematic review) would be limiting, but the way the paragraph is written, you are saying that a narrative review is limiting. Rewrite this paragraph to first say "This review is a narrative review" and cite a source explaining what narrative reviews are. Then say "We chose the narrative review format because…" or "A narrative review format was chosen because…" and explain why. Or something similar to this structure. Cite sources.
367 I don't think this counts as a "revelation" since it was already known before
368 If there is only one subsection, then there doesn't need to be a subsection. You do not need a heading for 2.1
378 The sentence about the oropharynx needs a citation
380 This paragraph needs citations
614-617 This feels redundant with earlier information. DOes it belong in this section?
622-625 This paragraph is redundant with text from earlier in this section. Delete it.
626+628 "Thus" and "as discussed above" are not correct here, because you have not provided any evidence for the claim in the rest of the sentence. Every paragraph before explained what the IM looks like, but you didn't discuss any of these envrionmental factors at all. I suggest you write a brand new section that only talks about these topics: a pararaph for diet citing at least 3 sources, a paragraph for xenobiotics citing at least 3 sources, a paragraph for drugs citing at least 3 sources, a paragraph for infections citing at least 3 sources.
628-633 What side effect? This text looks like it was taken from another manuscript. This text suggests the microbiome can alter the effect of digoxin, but I see no evidence that this is about dysbiosis. Delete it.
665-666 "Finally" is not right. This sentence also needs more explanation: expand it into a full paragraph and cite more sources. Also, why are you talking about apoptosis in a paragraph about anaerobic bacteria? Every paragraph needs to have one topic. Every sentence in that topic belongs in that paragraph.
697-706 This is a lot of text to describe one paper. Compared to the rest of the manuscript, it stands out as unusual. We do not need all of this information. Just say: "In a previous study on rates, a combination of glutamine and growth hormone reduced bacterial translocation over time in sepsis [69]." Then, add a few more single-sentence examples form the literature to support the statement made in 695-697.
752 Delete "Thus"
778 sentence fragment
799-809 Again, too much detail, when the rest of the paper lacked detail. For a review paper, you do not need this much detail. Instead, spend text on citing more examples.
829-843 These should be one paragraph, and could in fact be part of the preceeding paragraph. That way the connection between them all is clear.
921 Section 4.3 Is supposed to be about the role of paracellular permeability in Gi diseases. The sentences in the three paragraphs that follow do not seem to relate to paracellular permeability. The word "paracellular" does not even appear there. Upon re-reading I can see the connection, and maybe a reader familiar with the subject would see it, but it isn't obvious.
962 I do not follow the logic from the first sentences to the last. The sterility of 31% of nasogastirc aspirates tells us nothing about proximal intestinal colonization, bacterial translocation, or septic morbitity. The presence of E.coli in the samples also tells us nothing. "Thus" is not correct here, but also the purpose of this paragraph is unclear.
963-397 What does this have to do with "Role of sepsis in promoting intestinal bacterial translocation"?
976-979 There is no reason readers need to know these details. Delete.
980-989 What does this have to do with bacterial translocation?
1002&1010 You define bacterial translocation twice, separately, and in the middle of the section. It should be defined once in the beginning of the section [or once in the beginning of the paper, and then never defined again]. Stuff like this is why I believe the manuscript was written by multiple people and stiched together. A single author [probably] would not write this way.
1080-102 sentence fragment
1096 "mistreated" is probably the wrong word
1099-1113 I like this paragraph! Make the rest of the paper like this.
114-1136 What does this have to do with the gut-brain axis? It does not belong in this section.
1139-1155 Another example of a well-written paragraph.

Comments on the Quality of English Language

The language will need some editing, but overall cohesiveness is the bigger issue.

Author Response

Thank you for your insightful and helpful feedback. We have carefully considered your suggestions, along with those of the other four reviewers, and have incorporated the majority of them. For clarity, revisions made based on your comments are marked in purple, and deleted text is indicated in red.

While we appreciate your perspective, we note that the other four reviewers did not request a complete rewriting of the manuscript. We have strived to balance your feedback with the other reviewers' comments to achieve the best possible outcome for the manuscript. We believe this combined approach represents a significant improvement and hope it meets with your approval.

Round 3

Reviewer 5 Report

Comments and Suggestions for Authors

There are still issues with the paper including lack of cohesion. For example, the last paragraph of section 3, lines 275-283. The first sentence says changes in the IM can occur with exposure to various factors. The next two sentences describe an IM that can degrade digoxin, and the next describes a bacterial metabolite that affects drug metabolism. None of those are examples of the IM changing with exposure to drugs.

Comments on the Quality of English Language

Some typos scattered about that will hopefully be caught during proofreading, but nothing that interferes with my understanding of the text.